# An intercomparison of satellite, airborne, and ground-level observations with WRF-CAMx simulations of NO₂ columns over Houston, TX during the September 2021 TRACER-AQ campaign

M. Omar Nawaz[1], Jeremiah Johnson[2], Greg Yarwood[2], Benjamin de Foy[3], Laura Judd[4], and Daniel L. Goldberg[1]

[1]Department of Environmental and Occupational Health, George Washington University, Washington, DC, 20052, USA

[2]Ramboll, Novato, California, 94945, USA

[3]Saint Louis University, St. Louis, Missouri, 63103, USA

[4]NASA Langley, Hampton, Virginia, 23666, USA

*Correspondence to* M. Omar Nawaz (nawaz.muhammad@email.gwu.edu)

**Abstract.** Nitrogen dioxide ($NO_2$) is a precursor of ozone ($O_3$) and fine particulate matter ($PM_{2.5}$) – two pollutants that are above regulatory guidelines in many cities. Bringing urban areas into compliance of these regulatory standards motivates an understanding of the distribution and sources of $NO_2$ through observations and simulations. The TRACER-AQ campaign, conducted in Houston, TX in September 2021, provided a unique opportunity to compare observed $NO_2$ columns from ground-, airborne-, and satellite-based spectrometers. In this study, we investigate how these observational datasets compare, and simulate column $NO_2$ using WRF-CAMx with fine resolution (444 x 444 m$^2$) comparable to the airborne column measurements. We compare WRF-simulated meteorology to ground-level monitors and find good agreement. We find that observations from the GEOstationary Coastal and Air Pollution Events (GEO-CAPE) Airborne Simulator (GCAS) instrument were strongly correlated ($r^2$=0.79) to observations from Pandora spectrometers with a slight high bias (NMB=3.4%). Remote-sensing observations from the TROPOspheric Monitoring Instrument (TROPOMI) were generally well correlated with Pandora observations ($r^2$=0.73) with a negative bias (NMB=-22.8%). We intercompare different versions of TROPOMI data and find similar correlations across three versions but slightly different biases (from -22.8% in v2.4.0 to -18.2% in the NASA MINDS product). Compared to Pandora observations, the WRF-CAMx simulation had reduced correlation ($r^2$=0.34) and a low bias (-21.2%) over the entire study region. We find particularly poor agreement between simulated $NO_2$ columns and GCAS-observed $NO_2$ columns in downtown Houston an area of high population and roadway densities. These findings point to a potential underestimate of $NO_X$ emissions ($NO_X$ = NO + $NO_2$) from sources associated with the urban core of Houston, such as mobile sources, in the WRF-CAMx simulation driven by the Texas state inventory; and further investigation is recommended.

## 1 Introduction

Nitrogen dioxide ($NO_2$) is a critical precursor to criteria air pollutants (i.e., ozone or "$O_3$" and fine particulate matter or "$PM_{2.5}$") that are above regulatory thresholds in many urban areas. Exposure to $NO_2$ is also directly associated with asthma exacerbation in vulnerable groups (Achakulwisut et al., 2019; Anenberg et al., 2022) and premature death (Huang et al., 2021). Due to its short atmospheric lifetime (de Foy et al., 2014), observations of $NO_2$ can reveal fine-scale patterns associated with sources. A major source of $NO_2$ is fossil-fuel combustion (McDuffie et al., 2020) and in many urban airsheds this is the dominant contributor to $NO_2$; however, other natural sources – like lightning (Murray, 2016) and soil microbes (Hudman et al., 2012) – along with fires (Jin et al., 2021) and tropospheric-stratospheric $NO_2$ exchange also contribute to tropospheric $NO_2$ levels. The health burden, sources, and short atmospheric lifetime of $NO_2$ all compound in urban environments where there are large populations, diverse contributors, and unique fine-scale patterns in $NO_2$ levels.

In the US city of Houston, Texas – the fifth most populous metropolitan region in the US (United States Census Bureau, 2022) – $NO_2$ is a major concern (Mazzuca et al., 2016) due to its role as a precursor of the formation of $O_3$ and $PM_{2.5}$. While $NO_2$ itself nor $PM_{2.5}$ exceed their respective US EPA National Ambient Air Quality Standards (NAAQS), Houston is in moderate nonattainment of the 8-hour Ozone (2015) NAAQS. The large petrochemical industry in Houston emits $NO_2$ in addition to other common heavy emitting sources associated with coastal urban environments like vehicles, power stations, and shipping channels (Kim et al., 2011). The co-location of this large population with high levels of $NO_2$ presents a major public health concern that motivates research to better understand the sources that are most culpable in contributing to air pollution. Major highways like the I-610 interstate, the I-10 interstate, and Beltway 8 have heavy vehicle traffic that are responsible for elevated $NO_2$ concentrations (Miller et al., 2020). Large power stations and industrial facilities operate within and around the Houston metropolitan area and these point sources – along with a large shipping channel – are responsible for $NO_2$ plumes (Luke et al., 2010). Characterizing the unique imprints of these disparate sources remains a question of scientific concern. There is also evidence that low-income and non-white populations in Houston are disproportionally affected by air pollutants such as $NO_2$ (Demetillo et al., 2020).

Synchronous observations of $NO_2$ column densities from aircraft, ground-based, and satellite spectrometers coincided in September 2021 during the Tracking Aerosol Convection interactions ExpeRiment– Air Quality (TRACER-AQ). This campaign provided a unique opportunity to investigate the fine-scale patterns in $NO_2$ levels in Houston, TX. One of the devices employed during the TRACER-AQ campaign across its twelve flight days was the GEOstationary Coastal and Air Pollution Events (GEO-CAPE) Airborne Simulator (GCAS) instrument that has been discussed in many previous studies (e.g., Judd et al., 2020; Kowalewski & Janz, 2014; Leitch et al., 2014; Nowlan et al., 2018). The GCAS instrument is an ultraviolet-visible (UV-VIS) spectrometer. Its data is used to retrieve $NO_2$ columns over a limited number of flight days; this made its observational average more sensitive to meteorological conditions than an instrument with a longer time-record; however, this tool observes $NO_2$ pattens with uniquely fine-scale resolution (on average 560 x 250 $m^2$) and performed comprehensive measurements of $NO_2$

columns across large swaths of the city repeatedly up to three time per day. This differs with observations from the TROPOMI instrument on board the Copernicus Sentinel-5 Precursor (S5P) satellite that is in a near-polar sun synchronous orbit (van Geffen et al., 2022) that only observes $NO_2$ once per day in the early afternoon at a coarser resolution of 3.5 x 5.5 $km^2$ at nadir. TROPOMI and GCAS spectra are used to retrieve slant $NO_2$ columns that are converted into vertical columns using an air mass factor (AMF) (Palmer et al., 2001) which is the largest source of

uncertainty in the tropospheric vertical column retrieval algorithm (Lorente et al., 2019). Comparing TROPOMI data to other observations – like those from aircraft or ground-based monitors – can serve as a useful diagnostic in characterizing its performance and potential biases. These characterizations have large-scale implications since TROPOMI measures $NO_2$ columns globally and is useful in areas that lack the observational infrastructure of other instruments. The Pandonia Global Network (PGN) is a network of Pandora instruments (Herman et al., 2009); these

instruments are UV-VIS spectrometers that measure spectrally resolved radiance data that is used to retrieve total vertical $NO_2$ columns. A total of seven Pandora instruments were operational during the TRACER-AQ campaign across three separate sites in and around downtown Houston.

The Comprehensive Air Quality Model with Extensions (CAMx) is a multi-scale photochemical model that can

simulate air pollutants including ozone, fine particulate matter, and $NO_2$ (Ramboll, 2022b). CAMx has been used extensively to investigate Texas air quality by leveraging model input data created by the Texas Commission on Environmental Quality (TCEQ) for air quality planning (Ge et al., 2021; Goldberg et al., 2022) with strong performance compared to remote-sensing column concentrations in Texas (Goldberg et al., 2022; W. Li et al., 2023; Soleimanian et al., 2023). CAMx can be coupled with meteorological models like the Weather Research and

Forecasting Model (WRF) which provide the meteorological inputs necessary to simulate fine-scale atmospheric conditions (Jia et al., 2017) – this coupled modeling system is denoted as WRF-CAMx. Fine-scale simulations from WRF-CAMx are useful to understand biases in simulated $NO_2$ and to identify under- or over-estimates of emissions from sectors and regions in the inventories that drive the model.

In this study, we leverage the unique coincidence of ground-based Pandora spectrometers, high-resolution airborne, and TROPOMI-based remote sensing observations of column $NO_2$ during the September 2021 TRACER-AQ campaign (Judd et al., 2021). We assess the capabilities of these different data sources through cross-comparisons then compare observed $NO_2$ to simulated values from a WRF-CAMx simulation to evaluate its performance. Additionally, we consider the impact of different TROPOMI algorithms on performance against Pandora

measurements. Our comparisons across the three observational datasets clarifies the range of expected values of $NO_2$ column concentrations in Houston, TX and characterizes potential deficiencies and biases in observational products and simulated CAMx values. We investigate weekday-weekend performance of the model and consider differences in the spatial distributions of tropospheric $NO_2$ columns to qualitatively identify the sources that may be under- or overestimated in local inventories and identify the regions in Houston that are most likely impacted by

these incorrectly attributed emissions. Additionally, we compare diurnal profiles in column and surface concentrations of $NO_2$ across relevant products.

**2 Data and Methods**

**2.1 Pandora Observations**

During the TRACER-AQ campaign a total of seven Pandora instruments operated across three sites in Houston (Table 1). Pandora instruments are ground-based UV-VIS spectrometers that measure spectrally resolved radiances and this work only utilizes those collected via direct-sun observations (Herman et al., 2009). Trace gas spectral fitting routines are employed to characterize column concentrations of gases (e.g., $NO_2$) similar to remote-sensing and aircraft observations (Judd et al., 2020). Details on the Pandora instruments and their fitting routines are discussed in detail in past studies (Cede, 2021; Herman et al., 2009). The study was designed to have two Pandoras operating coincidently at each site during the campaign; however, due to instrument failures an uneven number of observations were obtained at each site. In order to evenly weigh the observations between the three sites, we select data from a single Pandora instrument at each site. Pan #58 at La Porte, #61 at Aldine, and #25 at University of Houston were chosen for the following reasons. As indicated in Table 1, Pan #61 and Pan #58 clearly have the largest temporal coverage during the TRACER-AQ time period. While Pan #188 measured more frequently at the University of Houston than Pan #25, Pan #188 was operated on a tower about 70 meters above the surface, which results in missing portions of the tropospheric column when operated in direct-sun mode.

Locations of the three sites are presented in Fig. 2F. These three chosen instruments are shaded and bolded in the table below. Pandora direct-sun retrievals represent the "total vertical column" of $NO_2$ which differs from the aircraft measurements that only measure the tropospheric column. We directly compare these disparate sources by adding a "stratospheric $NO_2$ column component" derived from TROPOMI estimates to the aircraft measurements (see section 2.2 and 2.3) to compare total column amounts.

**Table 1: Details on Pandora instrument operational time**

| Inst. # | Location | Lat. | Lon. | September 2021 Flight Days with Observations (Number of high- and medium-quality measurements per day) | | | | | | | | | |
|---|---|---|---|---|---|---|---|---|---|---|---|---|---|
| | | | | 1st | 3rd | 8th | 9th | 10th | 11th | 23rd | 24th | 25th | 26th |
| 11 | La Porte | 29.67 | -95.06 | 322 | 347 | 0 | 0 | 0 | 0 | 0 | 0 | 0 | 0 |
| **58** | **La Porte** | **29.67** | **-95.06** | **132** | **190** | **412** | **319** | **415** | **362** | **92** | **439** | **414** | **401** |
| 63 | La Porte | 29.67 | -95.06 | 0 | 0 | 0 | 0 | 0 | 0 | 0 | 0 | 265 | 207 |
| **61** | **Aldine** | **29.90** | **-95.33** | **168** | **253** | **400** | **391** | **419** | **367** | **420** | **420** | **405** | **420** |
| 148 | Aldine | 29.90 | -95.33 | 5 | 1 | 3 | 1 | 3 | 3 | 17 | 17 | 10 | 17 |
| **25** | **U of H** | **29.72** | **-95.34** | **213** | **256** | **300** | **299** | **392** | **273** | **400** | **382** | **400** | **344** |
| 188 | U of H | 29.72 | -95.34 | 528 | 610 | 1184 | 957 | 1137 | 722 | 372 | 749 | 225 | 95 |

**2.2 GCAS Observations**

The GCAS instrument was installed on the NASA G-V aircraft. The GCAS instrument employs charge-coupled device array detectors to observe backscattered light. These data can be used to retrieve column densities of gases like $NO_2$ below the aircraft using a DOAS computing software (Danckaert et al., 2017). During TRACER-AQ,
GCAS collected data over the Houston metropolitan area across 12 days during late August and throughout September 2021. The flight strategy of the aircraft included flying the plane in a 'lawnmower' fashion with flight lines spaced 6.3 km apart, ensuring overlap at flight altitude (FL280) with the instrument field of view of 45 degrees creating one gapless map of NO2 up to three times per flight day with an average differential slant column pixel size of 250 m × 250 m. $NO_2$ observations from GCAS are publicly available at the NASA Atmospheric Sciences Data
Center (NASA/LARC/SD/ASDC, 2022). Observations from two of the flight days – a test flight (August 30) and a flight over the Gulf of Mexico (September 27) are excluded from this study because they provided no meaningful data over Houston. Given the relatively short timeframe of flight data collection; meteorological conditions have an influence on the fine-scale patterns in $NO_2$ columns observations. Owing to this, we summarize some basic conditions and information of the 10 flight days that focused on Houston (Table 2). Wind and meteorological
conditions were determined by review of historical weather archives taken at Houston Hobby Airport (NASA, 2023; Weather Underground, 2023).

**Table 2: Basic meteorological conditions and notes during GCAS flights**

| Day of Sept 2021 | Day of the Week | High Temp | Wind direction | Additional note |
|---|---|---|---|---|
| 1 | Wed | 96 F | Weak SW winds | Thunderstorms from S to N, 11 AM to 4 PM |
| 3 | Fri | 93 F | Weak S winds | Scattered thunderstorms 12 PM to 4 PM |
| 8 | Wed | 94 F | N turning NE | Clear skies and no rain |
| 9 | Thurs | 95 F | N turning NE | Afternoon fair weather clouds, no rain |
| 10 | Fri | 93 F | NE turning E | Clear skies, no rain, some long-range smoke aloft |
| 11 | Sat | 93 F | E winds | Afternoon fair weather clouds, no rain |
| 23 | Thurs | 83 F | E winds | Clear skies, no rain, cold font overnight Sept 21 |
| 24 | Fri | 84 F | E turning SE | Clear skies, no rain |
| 25 | Sat | 87 F | NE turning E | Clear skies, no rain |
| 26 | Sun | 83 F | Calm then SE | Clear skies, afternoon fair weather clouds |


The publicly available GCAS measurements (version R2) include a version of the dataset with reprocessed AMFs to include $NO_2$ vertical profile estimates from the fine-scale ($444 \times 444$ m$^2$) WRF-CAMx simulation used in this analysis (section 2.4). Air mass factors use this vertical profile information to account for altitude-dependent sensitivities in remote-sensing observations. The original vertical profiles in the dataset were derived from a global model, GEOS-CF (Keller et al., 2021), that had a coarser spatial resolution ($0.25° \times 0.25°$). Lastly, to directly compare GCAS measurements to other $NO_2$ column concentrations we regrid them to a common grid; in this study, we chose the fine-scale WRF-CAMx grid. Only cloud-free GCAS data is considered in this analysis.

To characterize the accuracy and precision of GCAS measurements we compare them to observations from the Pandora instruments (section 3.1). This comparison requires both spatial and temporal screening. Spatially, we restrict our comparison to only the GCAS pixels that contain Pandora instruments. Temporally, we screen out all Pandora measurements that are more than 15 minutes removed from a GCAS overpass and then identify the Pandora measurement time within this 30-minute window that most closely matches the GCAS overpass time. While we choose this 30-minute window as an upper-bound cut-off, 96% and 90% of all Pandora closest matches occur within a 20- and 15-minute window of GCAS overpasses, respectively, indicating that this choice of window will have a minimal impact on our results. After screening the data, we also account for the fact that GCAS only measures the tropospheric component of the $NO_2$ column. There is a substantial but predictable "above-aircraft" column that is not reflected in the GCAS measurements. This is primarily associated with stratospheric $NO_2$. To account for this, we approximate the above aircraft component of the GCAS $NO_2$ columns using the stratospheric $NO_2$ column component of TROPOMI measurements (section 2.3) and add this to GCAS observations. Additionally, we add a "above aircraft" but below troposphere partial column amount based on the CAMx simulation; we calculate the column. In the 3 highest levels of CAMx that amounts to $0.57 \times 10^{15}$ molecules cm$^{-2}$ and add this amount to GCAS.

**2.3 TROPOMI Observations**

The TROPOMI instrument – on board the Sentinel-5P satellite – has measured total slant columns of $NO_2$ daily at approximately 13:30 local time globally from April 30, 2018 to present (European Space Agency, 2021). The slant column measurements were converted into tropospheric vertical column amounts by subtracting off a stratospheric $NO_2$ component and transforming the remaining tropospheric slant column to vertical column using an air mass factor. We download the publicly available data (https://data-portal.s5p-pal.com/products/no2.html; https://dataspace.copernicus.eu/) coincident with the TRACER-AQ campaign in September 2021 for overpasses of Houston, TX. In this study, we primarily consider measurements from the latest version (2.4.0) (Eskes et al., 2023); however, we additionally consider measurements processed using the version 2.3.1 algorithm (van Geffen et al., 2021) and the NASA Multi-Decadal Nitrogen Dioxide and Derived Products from Satellites (MINDS) product (Lamsal et al., 2022) and intercompare these different versions (Fig. 3). All product versions stem from the same slant column retrieval but differ in the calculation of the air mass factor for slant to vertical column conversions, and in the case of NASA MINDS, separation of the stratosphere and troposphere (Bucsela et al., 2013). The main

difference between 2.3.1 and 2.4.0 is the use of the 0.125° × 0.125° Directional Lambertian Equivalent Reflectivity (DLER) climatology derived from TROPOMI observations which replaces an old 0.5° × 0.5° Lambertian Equivalent Reflectivity (LER) dataset used in v2.3.1 (Eskes et al., 2023). NASA MINDS uses a geometry-dependent surface Lambertian Equivalent Reflectivity (GLER) product for their surface reflectivity input into the AMF calculation based on MODIS observations. The other main difference in these products include use of different a priori $NO_2$

profiles (1° × 1° TM5-MP for v2.3.1 and v2.4.0 vs 0.25° × 0.25° GMI simulation for NASA MINDS). A comparison between TROPOMI version 2.4.0 and a MAX-DOAS network found that in moderately polluted locations TROPOMI had a median bias of -35%; a comparison between TROPOMI version 2.4.0 and PGN found a median bias of -18% over polluted stations (Lambert et al., 2023).

These publicly available TROPOMI data are further processed for this study. We screen TROPOMI measurements

to consider cloud coverage and erroneous data using the recommended qa_value filter ($> 0.75$). We regrid the TROPOMI $NO_2$ observations (resolution of 3.5 x 5.5 $km^2$ at nadir) onto the WRF-CAMx grid (444 × 444 $m^2$). When comparing TROPOMI observations to Pandora instruments we follow the same spatial and temporal screening approach as discussed for GCAS. Spatially, we identify the CAMx grid cell in which each Pandora instrument is located and only consider TROPOMI measurements that were regridded to these grid cells. We

intercompare GCAS, TROPOMI, and CAMx at this resolution but also compare the three datasets at a coarser resolution (Section 3.4) to account for resolution-dependent errors. Temporally, we screen out all Pandora measurements that are more than 15 minutes removed from a TROPOMI overpass time and then identify the Pandora measurement time within this 30-minute window that most closely matches the TROPOMI overpass time. While we choose this 30-minute window as an upper-bound cut-off, 100% and 97% of all Pandora closest matches

occur within a 20- and 15-minute window of TROPOMI overpasses, respectively, indicating that this choice of window will have little impact on our results.  Using WRF-CAMx vertical profile information we calculate both a total and tropospheric $NO_2$ column from TROPOMI v2.4.0 measurements using new AMF derived from the WRF simulation and we difference the total and tropospheric values to calculate a stratospheric $NO_2$ column component from TROPOMI. We take the spatial and temporal average of this stratospheric component in Houston during the

TRACER-AQ campaign to calculate a constant bias correction to convert tropospheric $NO_2$ columns – from GCAS and WRF-CAMx – to quasi-total $NO_2$ columns when comparing them to total $NO_2$ column measurements from Pandora instruments. This stratospheric vertical column $NO_2$ amount of 3.0 ×$10^{15}$ is typical for Houston during summer (Geddes et al., 2018). Boersma et al. (2018) suggests that $0.5 × 10^{15}$ molecules $cm^{-2}$ is the upper limit of structural uncertainty in the stratospheric estimate; this uncertainty should be considered when reviewing results that

compare total column amounts (i.e., results comparing GCAS and CAMx to Pandora). We additionally account for diurnal variation in the stratospheric column by applying the results from work by K.-F. Li et al. (2021); they calculate a daytime stratospheric $NO_2$ column increase rate of $1.34× 10^{14}$ molecules $cm^{-2}$ starting at 7:00 LT. We apply this increase rate by calculating the difference in hours between the dataset times – either the GCAS overpass times or CAMx simulation hours – and 13:30 – the approximate TROPOMI overpass time – and then multiply this

difference by the increase rate. In doing so, total column values before the TROPOMI overpass are decreased and total column values after the overpass are increased.

**2.4 WRF-CAMx simulated NO$_2$**

For this study, a set of simulations were conducted employing version 4.3.3 of the Advanced Research Weather Research and Forecasting (WRF) model (Skamarock et al., 2021) jointly with the Comprehensive Air Quality Model

with Extensions (CAMx) v7.20 with the CB6r5 chemical mechanism for a simulation period that matched the September 2021 TRACER-AQ timeframe. A new high-resolution modeling platform was designed specifically for this study that adopted prior approaches used in Texas Commission on Environmental Quality (TCEQ) state implementation plan (SIP) modeling (TCEQ, 2021) to update emissions.

The WRF model is a mesoscale numerical weather prediction system designed to serve both operational forecasting and atmospheric research needs (Skamarock et al., 2005, 2008). We define the WRF modeling domains as slightly larger than the corresponding CAMx domains (Fig. 1) to avoid possible numerical artifacts near domain boundaries when transferring WRF meteorology to CAMx. The 36 km CAMx domain (red) includes the continental US, Mexico, and parts of Central America and Canada. The 36 km, 12 km (blue) and East Texas 4 km (green) domains

are also used by the TCEQ for State Implementation Plan (SIP) modeling. The higher resolution domains (1.333km (orange) and 0.444km (cyan)) were selected to include the most relevant GCAS flight tracks while considering computational expense.


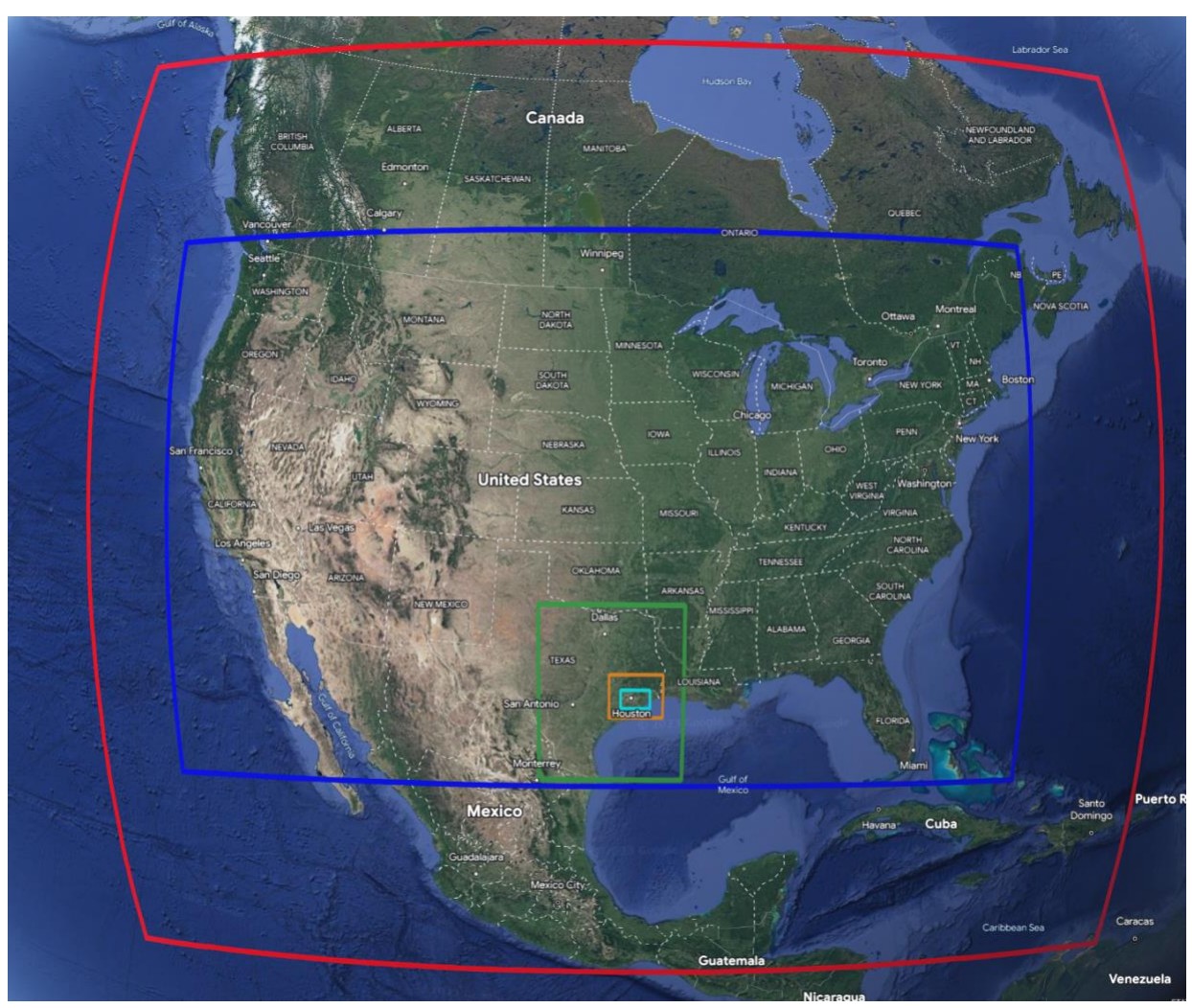

**Figure 1: Modeling domains used in the CAMx simulation for the 36 km resolution (red), 12 km resolution (blue), 4 km resolution (green), 1.333 km resolution (orange), and the 0.444 km resolution (cyan). Maps data provided by Google © 2020, Landsat / CopernicusData SIO, NOAA, U.S. Navy, NGA, GEBCO, IBCAO, INEGI, and U.S. Geological Survey.**

Additional information on the WRF-CAMx modeling is included in the supplemental including the WRF physics options (Table S1), vertical layer mapping from WRF to CAMx (Table S2), and CAMx science options (Table S3). We used 0.25° Global Forecasting System (GFS) data assimilation system (GDAS) analysis data (DOC/NOAA/NWS/NCEP/EMC, 2023) as initial conditions for the WRF meteorological model; this GDAS data is also used for boundary conditions and data assimilation. We configured the output timesteps of WRF to 15 minutes

for the higher resolution domains. Conducting WRF simulations at fine spatial resolutions (i.e., 4 km, 1.333 km, and 0.444 km) requires careful consideration of physical schemes that are sensitive to grid spacing. We turn off the convective cumulus parametrization scheme for the fine grids because WRF can explicitly simulate convection for them. For coarser grids we turn on the cumulus parametrization to account for sub grid-scale convection. The other

physics options (Table S1) are kept consistent across the different resolutions. The CAMx simulation was first
performed over the coarser domains (36 km, 12km, and 4 km) from which initial and boundary conditions were
extracted for the higher resolution domains. TCEQ developed the 2019 modeling emissions inventory for the Dallas-
Fort Worth (DFW) and Houston-Galveston-Brazoria (HGB) Attainment Demonstration (AD) SIP revisions (TCEQ,
2021). Starting with this inventory we implement further changes as discussed in the next paragraph.

First, we update the CAMx modeling emissions inventory from the TCEQ platform to incorporate 2021 hourly
Continuous Emissions Monitoring Systems (CEMS) (EPA, 2023) data for the eleven major electric generating units
(EGUs) listed in Table S4. We download hourly data from Clean Air Markets Program Data (CAMPD) for the
eleven EGUs for the August 30-Sep 27, 2021, period and stack parameters were based on the TCEQ 2019 emissions
platform (TCEQ, 2021). Second, we update shipping emissions to incorporate MARINER v2 (Ramboll, 2022a)
emissions built with 2021 Automatic Identification System (AIS) data for the higher resolution domains. Third, we
reprocess link-based on-road mobile emissions for the higher resolution domains. Fourth, we update biogenic
emissions and lightning $NO_X$ ($LNO_X$) based on WRF meteorology. Specifically, we use the Model of Emissions of
Gases and Aerosols from Nature (MEGAN) (Guenther et al., 2012) version 3.2 for biogenic emissions, the Fire
Inventory of NCAR (FINN) version 2.2 (Wiedinmyer et al., 2011) for fire emissions, and lightning $NO_X$ emissions
derived by applying the CAMx $LNO_X$ processor to the 2021 meteorological data from the WRF simulation.
Considering that wildfires in the Houston area are rare and that $LNO_X$ emissions are associated with convective
clouds that obscure remote sensing column observations, we excluded these two emission sources from the finer
resolution domains (the 1.333 and 0.444 km domains) but included them in the larger domains. These two sources
represent a small fraction of emissions in the local Houston area that is the primary focus of the finer resolution
simulations. Lastly, we regrid all other gridded emissions from the coarser domains to the high-resolution domains
without refining their spatial resolution. Specifically, all point sources are geo-located to the grid cell containing the
source. On-road mobile source emissions and shipping emissions were provided for individual links which we
allocated to 444 m grid cells, and are based on known roadway networks, ship tracks, and traffic patterns. Airport
and railyard emissions were allocated to 444 m grid cells within the property boundary. Other sources retained the 4
km grid resolution provided by the TCEQ. Daily emissions of $NO_X$ and volatile organic compounds (VOCs) in tons
per day (tpd) for a September weekday are presented in Table 3 below.

We evaluate the WRF simulation meteorology by comparing surface-level wind speed, direction, temperature, and
water vapor mixing ratio to observations from sixteen ground-level monitors (Table S5-8 and Fig. S2-11) and
calculate the mean bias error (MBE), mean absolute error (MAE), and Pearson-R squared ($R^2$) statistics as defined
in supplement table S2. Circular statistics are calculated using the Astropy circular statistics module for Python (The
Astropy Collaboration, 2022). We obtain integrated surface data from NCDC in the DS3505 format
(ftp://ftp.ncdc.noaa.gov/pub/data/noaa/); these data consist mainly of airport locations and have good meteorological
siting and quality assurance procedures.

Generally, meteorological conditions simulated by WRF agree with ground-level observations especially on the more data rich non-cloudy days that are the most important for our intercomparison; however, performance depends on the specific measure of meteorology considered. Across all days, the WRF wind direction was well correlated ($R^2$=0.76) and had minimal bias (MBE=8°), but some unsystematic errors (MAE=26°) compared to observations. This indicates that the model generally captures variability in wind direction without a notable bias; however, considering any individual observation the simulated direction may differ by 20°-30°. For non-cloudy days – that are more relevant for our intercomparisons due to more data – correlation for wind direction was similar ($R^2$=0.73) and the bias and error were reduced (MBE=-5° and MAE=21°). Simulations of wind speed were more poorly correlated ($R^2$=0.26) and had some unsystematic error (MAE=1.20 m/s); however, there was very little systematic bias in the wind speed simulation (MBE=-0.02 m/s). Correlation and unsystematic errors improve on the non-cloudy days ($R^2$=0.37 and MAE=1.08) while there is still no notable systematic bias (MBE=-0.13 m/s). Considering wind speeds at 9am and 1pm (Fig. S2-11), it appears that observations in the afternoon degrade correlation compared to the morning and that, generally, simulated wind speeds are better correlated with observations in downtown Houston than in the south-eastern part of the domain near the Galveston Bay. Comparisons between GCAS observations and WRF-CAMx simulations show that the model represents the dominant direction and dispersion of identifiable plumes from known sources. The wind speed bias is sufficiently low that model uncertainty will not lead to systematic errors in plume advection. Across the eight non-cloudy days, hourly and site-specific – across the sixteen monitors – WRF wind direction ($R^2$=0.3 to 0.8; MAE=14° to 32°), wind speed ($R^2$=0.1 to 0.5; MAE=0.94 m/s to 1.35 m/s), temperature ($R^2$=0.69 to 0.81; MAE=0.93 K to 1.18 K), and water vapor mixing ratio ($R^2$=0.28 to 0.78; MAE=0.87 g/kg to 3.11 g/kg) performed moderately compared to observations given the fine spatial and temporal resolution. Additionally, we compare simulated hourly $NO_2$ (Fig. S12) and maximum daily eight-hour average or "MDA8" $O_3$ (Fig. S13) to observations from seventeen TCEQ continuous air monitoring stations (CAMS) operating in Houston. We find poor performance and a strong negative bias in the simulated surface-level $NO_2$ (NMB=-59%) while simulated surface-level MDA8 $O_3$ has a much weaker bias (NMB=-2.5%) compared to observations. Comparisons to ozonesondes (Fig. S14-S18) suggest that WRF simulates more aggressive vertical mixing than what is observed; this is consistent with our findings of a stronger negative bias at the surface-level than for the columns as emitted $NO_2$ at the surface is advected vertically quicker in WRF-CAMx than in reality.

**Table 3: CAMx 444 × 444 m$^2$ domain-wide summary of average September weekday emissions by sector in units of tons per day (tpd).**

| Emission Sector | Spatial Resolution | NO$_X$ (tpd) | VOC (tpd) |
|---|---|---|---|
| EGUs | Point locations | 25.5 | 0.2 |
| On-road mobile | Line source | 70.9 | 34.7 |
| Railyards | 444 m gridded | 4.2 | 0.3 |
| Shipping | Line source | 63.9 | 4.3 |
| KIAH airport | 444 m gridded | 6.4 | 0.8 |
| KHOU airport | 444 m gridded | 1.8 | 0.4 |
| Other | | | |
|    Off-road mobile | 4 km gridded | 33.1 | 31.4 |
|    Non-EGU Point Sources | Point locations | 47.9 | 27.8 |
|    Oil and Gas | 4 km gridded | 0.2 | 0.0 |
|    Area | 4 km gridded | 92.8 | 623.2 |
|    MEGAN biogenic | 444 m gridded | 25.9 | 319.7 |

**2.5 Diurnal Comparison**

We further intercompare these data by grouping them at locations and then calculating their average diurnal profiles during the TRACER-AQ campaign for both column and surface-level NO$_2$. Specifically we compare GCAS, CAMx, Pandora, and GEOS-CF (Keller et al., 2021) NO$_2$ columns at the three Pandora sites during TRACER-AQ flight days. We include NO$_2$ data from GEOS-CF – that will be used for processing NO$_2$ remote-sensing observations from the NASA Tropospheric Emissions: Monitoring of Pollution (TEMPO) mission – to characterize

differences between a global simulation and our regional WRF-CAMx modeling. Simulated surface and NO$_2$ columns from GEOS-CF are obtained through the GMAO OPeNDAP interface (https://opendap.nccs.nasa.gov/dods/) for all of 2021 and filtered to the specific Pandora instrument locations and during TRACER-AQ flight days. We apply both spatial and temporal screening. Spatially, we identify the CAMx grid cell – for GCAS and CAMx – and GEOS-CF grid cell in which the Pandora instrument is located. Temporally,

for GCAS, we round all overpass times to the nearest hour and calculate the median value for each hour across all overpasses and days. For CAMx, GEOS-CF, and Pandora we identify the simulated and observed NO$_2$ column concentration closest to the hour and calculate the median value across all flight days and locations.

    For diurnal comparisons at the surface, we use surface-level NO$_2$ concentrations from CAMx and GEOS-CF and apply the same temporal screening. Spatially, for the surface-level we consider concentrations at a point in between

the three Pandora instruments that is representative of downtown concentrations (29.7 °N, 95.3 °W). We choose this point to represent the temporal behavior of the wider regions rather than individual sites. Additionally, we download

hourly NO$_2$ concentrations from the US Environmental Protection Agency (EPA) Air Quality System (AQS) (https://aqs.epa.gov/aqsweb/airdata/download_files.html). We download all hourly data for 2021 for the US and filter the TRACER-AQ flight days and for monitors in Harris County. We identify the median hourly concentrations across these monitors and the TRACER-AQ flight days.

## 3 Results

### 3.1 Comparisons to Pandora Observations

The observations from ground-based Pandora instruments are considered the most accurate of all observational platforms measuring column NO$_2$ presented in this project due to low uncertainties in their air mass factors (Herman et al., 2009) when operating in direct-sun mode. The air mass factor in this mode is calculated from simple solar geometry – unlike TROPOMI and GCAS, which rely on a priori assumptions like the vertical NO$_2$ profile and surface reflectivity. Pandora AMFs are not reliant on an a priori profile as the data we are using is only in direct-sun mode in cloud-free scenes. AMF for Pandora is analogous to pathlength through the atmosphere relative to the vertical path. Since all the signal is from direct sun path (extremely minimal scattering), this is purely geometric. Given this, we use Pandora observations as our reference dataset to characterize the performance of the two observational datasets – GCAS and TROPOMI – along with the WRF-CAMx simulation across three sites (Table 1). These three sites (Aldine, La Porte, and University of Houston) are located in the heavily polluted inner region of Houston that we denote as "urban Houston" (Fig. 2F). Background observations from Pandora instruments in less polluted sites were unavailable during the TRACER-AQ campaign so there is less certainty about the performance of GCAS, TROPOMI, and CAMx outside of urban Houston. We consider the performance of GCAS processed with a CAMx-based AMF (Fig. 2A), TROPOMI processed with a CAMx-based AMF (Fig. 2B), and CAMx (Fig. 2C) and the performance of GCAS and TROPOMI with the operational AMFs (Fig. 2D and 2E) individually and then intercompare the three datasets across the ten GCAS observation days (Fig. 2G).

When comparing the observational and simulated datasets to Pandora observations we consider the total column NO$_2$ – we add a stratospheric component from TROPOMI to the tropospheric column NO$_2$ of GCAS and CAMx to total column as discussed in the methodology. For TROPOMI, we use an AMF derived from the CAMx simulation to calculate a tropospheric NO$_2$ column from TROPOMI; following the TROPOMI users guide, we multiply the total averaging kernel by the ratio of the total air mass factor to the tropospheric air mass factor. We difference the total column NO$_2$ from TROPOMI with the tropospheric column to estimate a constant stratospheric NO$_2$ column amount that we add to GCAS and CAMx when we compare them to Pandora; this corresponds to a mean value of $3.0 \times 10^{15}$. For GCAS, we apply an additional amount to account for the NO$_2$ column in the upper troposphere – above the aircraft and below the tropopause. We calculate the column of levels 27-29 that correspond to 9400-18100 m above sea-level – that extends roughly from the height that GCAS flies at of around 9100 m to the tropopause – and apply this to the GCAS results; this corresponds to a value of $0.57 \times 10^{15}$ molecules cm$^{-2}$. For all results that

include comparison to Pandora we present total $NO_2$ columns; for results where we only intercompare GCAS, TROPOMI, and CAMx we compare the tropospheric column. All statistical measures (e.g., $R^2$) are defined in the supplement.

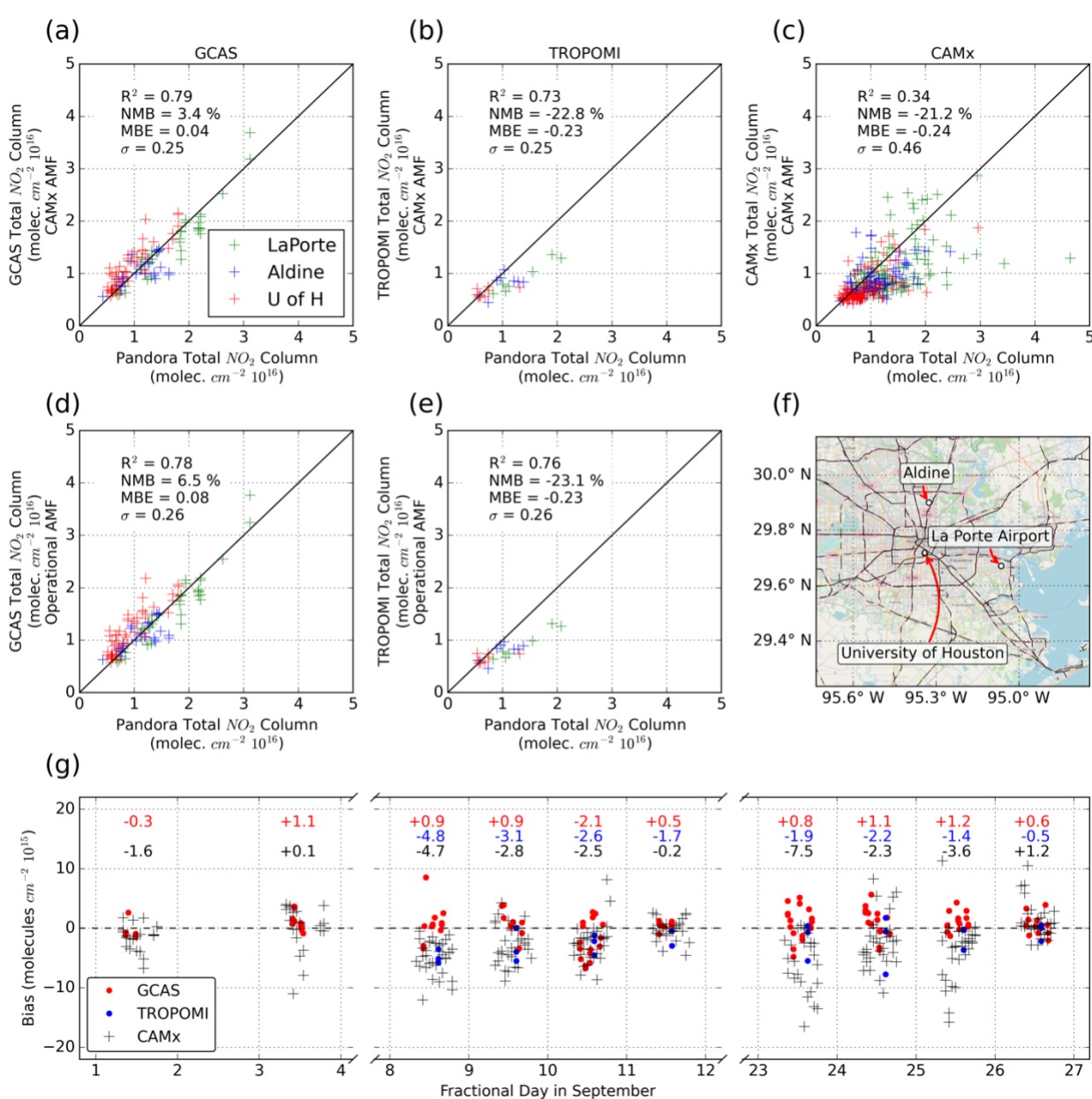


**Figure 2: Comparison of Pandora total column NO₂ to GCAS using CAMx-based AMFs (A), TROPOMI v2.4.0 using CAMx-based AMFs (B), and CAMx (C) and GCAS (D) and TROMPOMI v2.4.0 (E) with their operational AMFs. Tropospheric columns from GCAS and CAMx are bias corrected with a TROPOMI-derived stratospheric column factor as discussed in the methodology. Data from all possible overpasses**



In Fig 2A-C we characterize the performance of the observational and simulated datasets of $NO_2$ column concentrations across the three sites in Houston. For each of the GCAS flight days, we compare GCAS and TROPOMI observation against Pandora measurements for every overpass that was not obstructed by cloud coverage; for CAMx we compare simulated columns for every daytime hour of each GCAS flight day.


Observations from GCAS were both well correlated ($r^2$=0.79) and slightly high biased (NMB=+3.4%) when compared to measurements from Pandora. Use of the CAMx AMF in place of the operational AMF had a minimal impact on comparisons to Pandora (from $r^2$=0.78 and NMB=+6.5%). Observations from TROPOMI on GCAS flight days were also well correlated with Pandora measurements ($r^2$=0.73) but there was a negative bias (NMB=-22.8%) in v2.4.0. This bias was worse for more $NO_2$ polluted scenes. This negative bias may be attributable to the coarser resolution of TROPOMI compared to GCAS that weakens its ability to capture fine-scale plumes (Wagner et al., 2023) of $NO_2$ associated with road systems, airports, power stations, and industrial facilities. Similar to GCAS, use of the CAMx AMF in place of the operational AMF for TROPOMI had a minimal impact on comparisons to Pandora (from $r^2$=0.76 and NMB=-23.1%).



We calculate the ratios of the TROPOMI v2.4.0 product with the CAMx AMF compared to the operational AMF (Fig. S1) in September 2021 throughout the domain and note that tropospheric column $NO_2$ increases in the urban core and decreases in the city outskirts. The areas with Pandora instruments – in suburban Houston – have roughly equivalent values. Given that Pandora instruments were not located at either the most or least polluted areas of the metropolitan area, the benefit of the CAMx AMF may be underrepresented by our findings at the Pandora sites.


We compare simulated $NO_2$ columns from CAMx with Pandora measurements; however, in this comparison there are more points to intercompare as columns were simulated for each hour of every flight day by CAMx and observed multiple times per hour from Pandora. The CAMx simulated columns were less correlated with Pandora measurements ($r^2$=0.34) than compared to TROPOMI and GCAS, and they had a consistent negative bias (NMB=-21.2%). This poor correlation could partially be explained by differences in WRF simulated meteorology and observed meteorology specifically from differences in wind speed and direction and an inability to fully capture the bay breeze in Houston. We find that the WRF simulated wind direction ($R^2$=0.76 and MBE=8°), temperature



($R^2$=0.71 and MBE=0.39K), and water vapor mixing ratio ($R^2$=0.86 and MBE=-1.45 g/kg) (Table S5-S8 and Fig. S2-11) are generally well correlated and minimally-biased compared to observations; however, there are some unsystematic errors in wind direction (MAE=26°) and poor correlation in wind speed ($R^2$=0.26) that would likely degrade correlation between observed and simulated $NO_2$ columns. While there are errors in the meteorological conditions, the biases at the surface are all small – including minimal bias in the wind speed (MBE=-0.02 m/s) – indicating that the negative biases in $NO_2$ columns are likely attributable to an underestimate of $NO_x$ emissions; however, the WRF meteorological performance could partially explain the poor correlation and absolute errors in simulated $NO_2$ columns. We also note that generally, the model performance is stronger on windier days – when speeds exceed 4 m/s ($R^2$=0.5 and 0.32) – than on calmer days – when speeds are below 3 m/s ($R^2$=0.07, 0.1, and 0.25). Additionally, there can be substantial differences in vertical mixing coefficients in different schemes in the models, and these can impact the biases in column concentrations (de Foy et al., 2007; Riess et al., 2023). We briefly compare meteorology and the ozone mixing ratio in the WRF-CAMx simulation to ozonesondes data (https://www-air.larc.nasa.gov/cgi-bin/ArcView/traceraq.2021) and find that while temperature and pressure are captured well, there is variable performance in the vertical structure for ozone mixing ratio, wind speed, and wind direction (Fig. S14-S18).

In Fig. 2G, we intercompare the daily variability in biases across the ten GCAS flight days. There were no TROPOMI data for the first two flight days because cloud coverage blocked TROPOMI observations at the Pandora sites during its overpass time. The daily average bias of GCAS observations were consistently small throughout the entire period: they ranged from -2.1 to +1.2 molecules cm$^{-2}$ 10$^{15}$ on September 10[th] and the 3[rd] and 24[th], respectively. TROPOMI observations were consistently biased systematically low: they ranged from -4.8 to -0.5 molecules cm$^{-2}$ 10$^{15}$; however, on all days except the 26[th] daily averaged TROPOMI biases were more negatively biased than -1.3 molecules cm$^{-2}$ 10$^{15}$ compared to Pandora measurements. Unlike the two observational datasets, the bias in simulated CAMx $NO_2$ columns had much higher daily variability. On some days, such as September 3[rd], there was little bias in simulated columns compared to Pandora measurements, and on other days such as September 26[th], there was a minor high bias (+1.2 molecules cm$^{-2}$ 10$^{15}$); however, on most days there was a negative bias that was the strongest on September 23[rd] when $NO_2$ columns were biased as low as -7.5 molecules cm$^{-2}$ 10$^{15}$. Generally, simulated CAMx columns perform better on weekend days (11[th], 25[th], and 26[th]) which is investigated in greater detail in section 3.4.

**3.2 Comparisons of different TROPOMI algorithms to Pandora Observations**

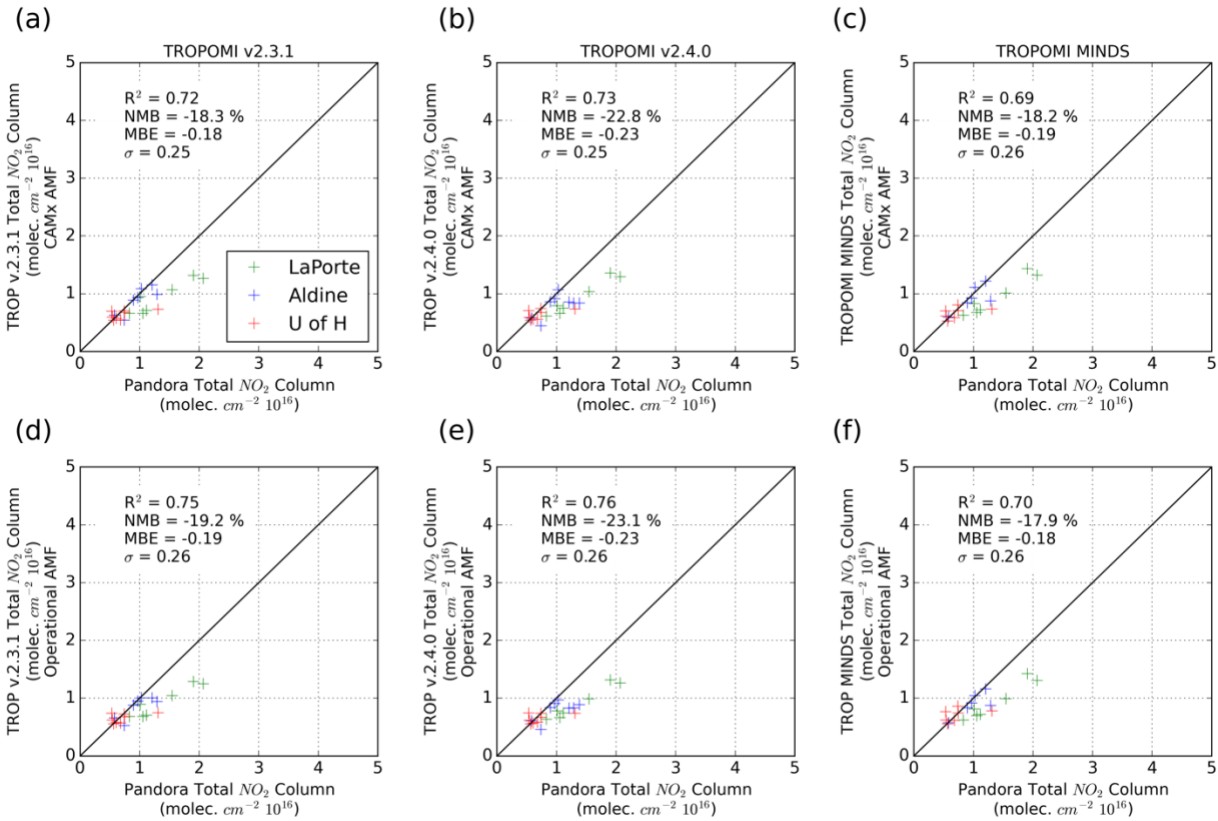

**Figure 3: Comparison between Pandora measurements and TROPOMI observations using the CAMx AMF for version 2.3.1 (A), 2.4.0 (B), and NASA MINDS (C) and the same respective versions using the operational AMF (D-F). Data from all possible overpasses coincident within 15 minutes of a Pandora observation are considered with one exception: data from September 11th, 2021, was missing from the NASA MINDS product and so values in plots C and F exclude this day.**

We intercompare TROPOMI observations to Pandora measurements across three different algorithms: version 2.3.1 (Fig. 3 A, D), version 2.4.0 (Fig. 3B, E), and the NASA MINDS product (Fig. 3C, F) using both the CAMx AMF (top row) and the Operational AMF (bottom row) for the same Pandora instruments in Houston during the TRACER-AQ Campaign. Overall, the choice of algorithm and AMF does affect the performance of TROPOMI compared to Pandora, albeit slightly. Regardless of AMF, version 2.4.0 appears to have the worst normalized mean bias in Houston during TRACER-AQ ($r^2$=0.73, and NMB=-22.8%), version 2.3.1 is improved ($r^2$=0.72 and NMB=-18.3%) while the NASA MINDS product performs comparably ($r^2$=0.69 and NMB=-18.2%) to version 2.3.1. Notably, NASA MINDS data for September 11th are missing so these data are excluded from panels C and F. For version 2.3.1 and version 2.4.0 the CAMx AMF slightly improves the bias; however, for the MINDS product the CAMx AMF slightly worsened bias compared to the operational AMF. The correlation is generally unaffected by the

choice of AMF. We choose TROPOMI version 2.4.0 for the intercomparison in the following sections as it is the most recent version.

### 3.3 Comparisons of GCAS, TROPOMI, and CAMx data on the CAMx grid

The comparisons between Pandora measurements and the datasets indicate that GCAS observations are in best agreement with Pandora. While TROPOMI performs worse than GCAS, it still decisively outperforms simulated

$NO_2$ columns from CAMx at Pandora sites in both correlation and bias despite its coarser resolution. With the above in mind, in this section we present $NO_2$ columns observed from GCAS and TROPOMI and simulated from CAMx at the $444 \times 444$ m$^2$ resolution of the CAMx grid. We extend the prior comparison beyond focusing on three discrete points in urban Houston to the entire CAMx domain to get a more complete picture of the spatial components of these datasets. For each dataset we consider observations across all ten GCAS flight days. We begin by comparing

GCAS observations only with CAMx simulated columns across all GCAS overpasses as these data are less limited temporally than TROPOMI observations (Fig. 4).

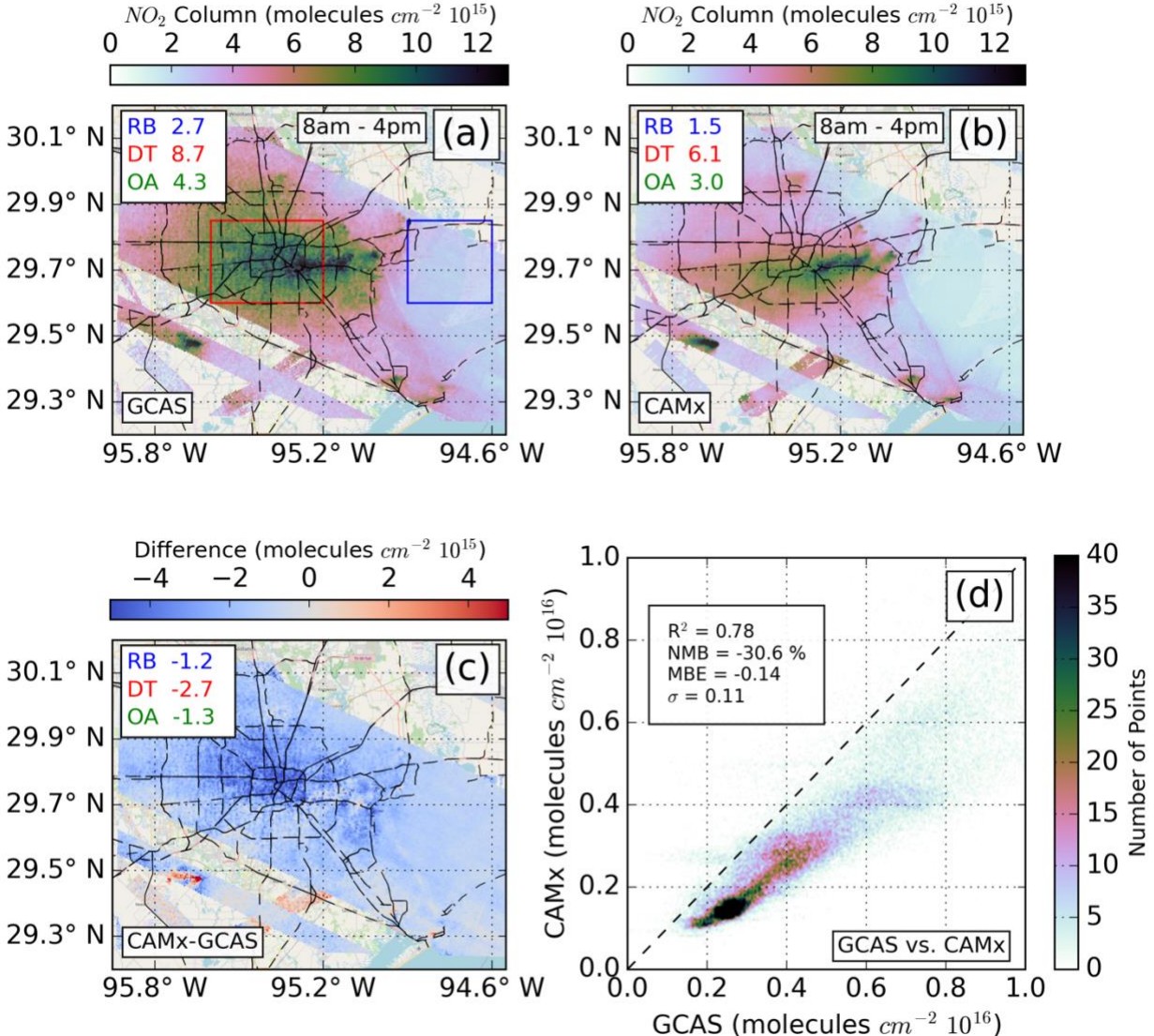

Figure 4: Comparison of GCAS observations to CAMx simulated NO₂ columns across all data during GCAS overpasses (generally 8 am – 4 pm). Temporally averaged GCAS NO₂ columns (A), temporally averaged simulated CAMx NO₂ columns (B), the absolute difference between GCAS and CAMx (C), and a scatter density plot comparing all observations between GCAS and CAMx (D). We identify three distinct areas: downtown or "DT" (red), the low emissions East Galveston rural Bay or "RB" (blue), and all other areas or "OA" (green) and calculate the averages in the top left of each chart. © OpenStreetMap contributors *2023*. Distributed under the Open Data Commons Open Database License (ODbL) v1.0.

When considering data from all GCAS overpasses (Fig. 4A, B, C) we observe a consistent negative bias in the CAMx product compared to GCAS observations throughout the domain that worsens in the downtown (DT) area (-2.7

molecules cm$^{-2}$ 10$^{15}$) compared to background levels in the rural East Galveston Bay (RB) (-1.2 molecules cm$^{-2}$ 10$^{15}$). Near the W A Parish power station in the southwestern area of the domain there are a mixture of positive and negative biases in the CAMx simulated columns that are likely indicative of errors in wind speeds or directions in the CAMx simulation. Overall, the CAMx simulated columns were well correlated with GCAS observations (r$^2$=0.78) but the

negative bias was substantial (NMB=-30.6%) (Fig. 4D).

We continue this comparison in Fig. 5 where we limit the GCAS and CAMx values temporally around TROPOMI overpasses. For Fig. 5, we screen out all observations that are +/- 90 minutes from TROPOMI overpass for each day and then temporally average the observations across the GCAS flight days (Fig. 5A-C). We difference, both

absolutely (Fig. 5D-F) and relatively (Fig. 5G-I), the three pairs of datasets and present them in scatter density plots (Fig. 5J-L). We focus on three regions: downtown Houston (DT) (red), the rural East Galveston Bay (RB) (blue), and all other areas (OA) (green) and calculate the mean values and differences for these areas in the top left of each of the plots. The results presented in Fig. 5 are the temporal average across all flight days; however, similar figures for individual flight days are presented in the supplemental (Fig. S19-28)


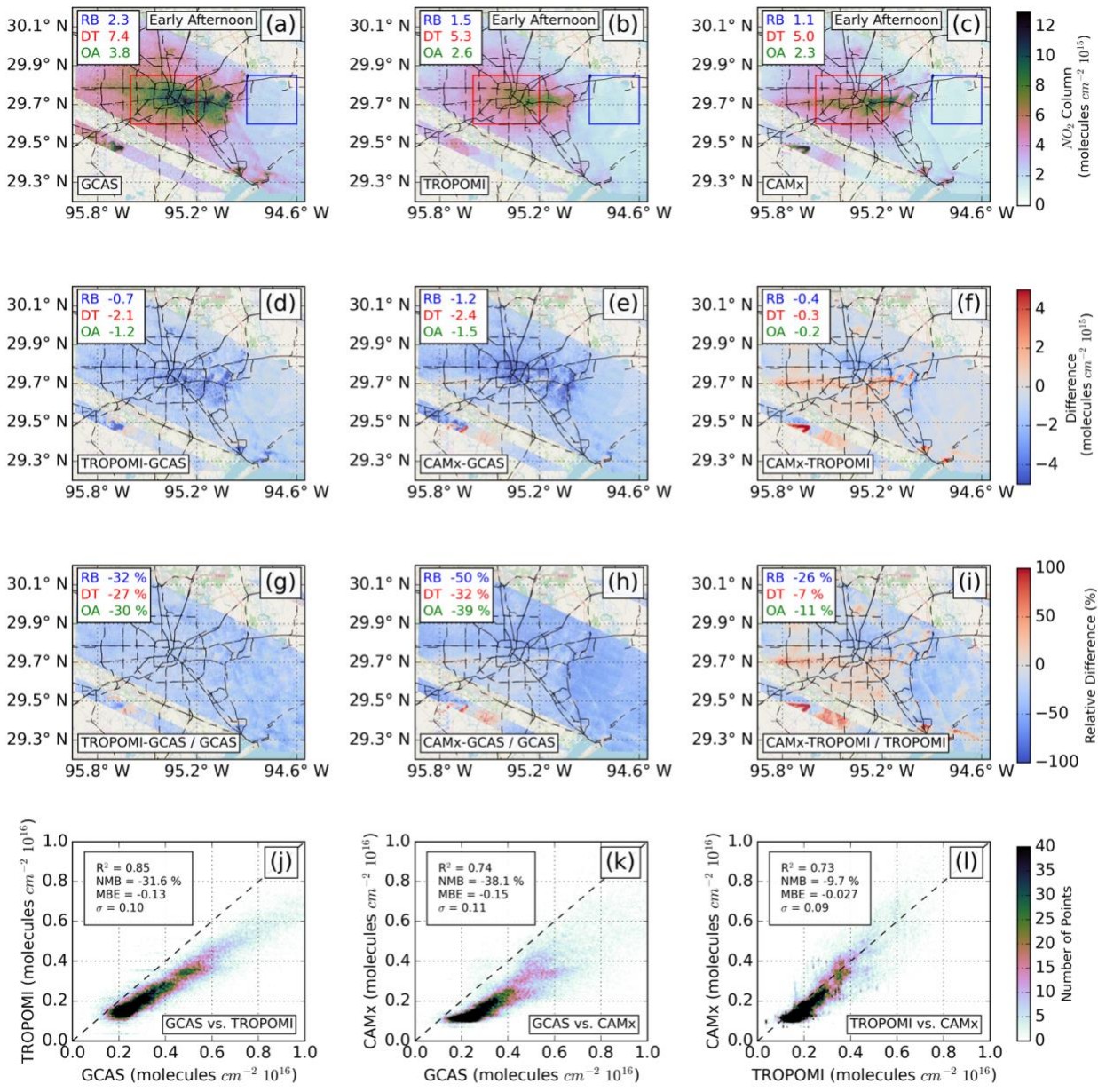

**Figure 5: Spatial distribution of GCAS (A), TROPOMI (B), and CAMx (C) NO₂ columns averaged across the ten GCAS flight days when within 90 minutes of each TROPOMI overpass representing early afternoon NO₂ columns. We identify three distinct areas: downtown or "DT" (red), the low emissions East Galveston Bay or "RB" (blue), and all other areas or "OA" (green) and calculate the averages in the top left of each chart. Absolute differences between GCAS and TROPOMI (D), GCAS and CAMx (E), and TROPOMI and CAMx (F). Relative differences between GCAS and TROPOMI (G), GCAS and CAMx (H), and TROPOMI and CAMx (I). Scatter density plots of GCAS vs. TROPOMI (J), GCAS vs. CAMx (K), and TROPOMI vs. CAMx (L). © OpenStreetMap contributors 2023. Distributed under the Open Data Commons Open Database License (ODbL) v1.0.**

First, we consider the spatial distribution of $NO_2$ columns from GCAS (Fig. 5A), TROPOMI (Fig. 5B), and CAMx (Fig. 5C) independently. For all three datasets, $NO_2$ columns are higher in downtown Houston than in the rural East Galveston Bay; generally, they are between 3 and 5 times as large. The two finer-resolution datasets – GCAS and CAMx – also capture $NO_2$ peaks associated with point sources like those from W A Parish, Texas City, and Baytown and in the Ship Channel. A map of the major point sources discussed in this work is included in the supplemental (Fig. S29). The coarser resolution of TROPOMI leads to fewer identifiable peaks associated with point sources; however, there are slightly elevated observed values near the W A Parish and Texas City power plants and the Ship Channel. Observations from GCAS and TROPOMI reveal a more diffuse peak in $NO_2$ columns in and around downtown Houston that includes elevated levels of $NO_2$ in the western part of the city. Simulated columns from CAMx, on the other hand, primarily estimate higher $NO_2$ values in the eastern area of downtown Houston and have lower $NO_2$ values in the western area of the city.

We next consider the three products compared to one another through three methods: absolute difference (Fig. 5D-F), relative difference (Fig. 5G-I), and scatter density plots (Fig. 5J-L). We intercompare these three products by isolating three sets of pairs: that is GCAS and TROPOMI, GCAS and CAMx, and TROPOMI and CAMx.

First, considering GCAS and TROPOMI, there appears to be a systematic low bias in TROPOMI observations throughout nearly the entire domain. Regardless of the spatial subset, the low bias in TROPOMI was consistent and ranged from -27% in downtown to -32% in the rural bay (Fig. 5G). In an absolute sense, on average TROPOMI was between 2.1 and 0.7 molecules cm$^{-2}$ 10$^{15}$ lower than GCAS (Fig. 5D) across the three locations. Throughout the entire domain, observations from GCAS and TROPOMI were well correlated ($r^2$=0.85), but TROPOMI had an overall negative normalized mean bias of -31.6% (Fig. 5J). We note that this low bias is slightly greater than what we would expect from considering the biases of these products relative to Pandora measurements as we do in section 3.1; doing this we would expect TROPOMI to be low biased relative to GCAS by around 23%. This slight additional negative bias indicates that either the three Pandora sites are unable to capture the full extent of the negative TROPOMI bias and that TROPOMI may be lower biased outside of these sites (e.g., areas outside of downtown Houston) or that GCAS observations may be biased additionally high outside of these sites. Notably, there are a few areas surrounding point sources in the eastern area of downtown and around the W A Parish plant in which TROPOMI observes higher $NO_2$ columns than GCAS. This is likely attributable to the coarser resolution of TROPOMI that results in peaks of $NO_2$ to be spread into surrounding areas that are in the same TROPOMI grid cell.

Second, comparing GCAS to CAMx we again find a low bias relative to GCAS, albeit one with a higher degree of spatial variability. In the remote bay, CAMx simulated columns are lower than GCAS compared to elsewhere in the domain (-50%) (Fig. 5H) while downtown and background levels are similarly biased at 32% and 39%, respectively. This lower bias in the low emission East Galveston Bay is indicative of an underestimation of background $NO_2$ columns in the CAMx simulation. Across these three regions the mean absolute differences range from -2.4 to -1.2

molecules cm$^{-2}$ $10^{15}$ (Fig. 5E). Visually, the negative bias in CAMx appears to be stronger in downtown and to the west, east, and north-west of downtown and less to the south and south-west of downtown. Overall, GCAS and CAMx are well correlated ($r^2$=0.74) (Fig. 5K); however, simulated columns from CAMx have a worse negative bias (NMB=-38.1%) against GCAS than what is captured at the Pandora sites of approximately -21%. Around some point sources CAMx columns are positively biased against GCAS observations. This high bias in CAMx is likely attributable to differences in wind speed and direction in the WRF simulation than in reality. These differences could contribute to NO$_2$ plumes being advected in incorrect directions.

Lastly, when comparing observed columns from TROPOMI to simulated columns from CAMx, biases have a great degree of spatial variability; however, in general CAMx is negatively biased. In a relative sense (Fig. 5I), the CAMx simulated columns are lowest compared to TROPOMI in the rural bay (-26%) and similar in downtown (-7%) and in other areas (-11%). There are a few areas where this pattern does not hold: both in the area southwest of downtown Houston and near point sources, CAMx is biased high compared to TROPOMI. These results indicate that simulated columns from CAMx are underestimated in downtown Houston and that this underestimation could potentially be attributable to an incorrect advection of NO$_2$ from some downtown source to the south-west perhaps in conjunction with an underestimate of emissions in this downtown area. Overall, TROPOMI and CAMx are well correlated ($r^2$=0.73) and there is a spatially heterogeneous low bias when considering the two products throughout the domain (NMB = -9.7%) (Fig. 5L).

### 3.4 Comparisons of GCAS, TROPOMI, and CAMx data at a coarser resolution

The comparisons presented in the prior section are done at the high resolution of the CAMx grid (444 × 444 m$^2$). Here, we characterize the effect of the coarser resolution of TROPOMI by performing an additional comparison of the three datasets at the 0.05° × 0.05° resolution (approximately 5.5 × 5.5 km$^2$) (Fig. 6). We average all of the NO$_2$ columns from this finer resolution to the coarser resolution based on the centroid of the fine resolution grid cells. This new coarser resolution is comparable to that of the TROPOMI observations at nadir (on average 3.5 × 5.5 km$^2$). We additionally present comparisons at two further coarser resolutions in the supplemental: 0.25° × 0.25° (Fig. S30) and 0.1° × 0.1° (Fig. S31).

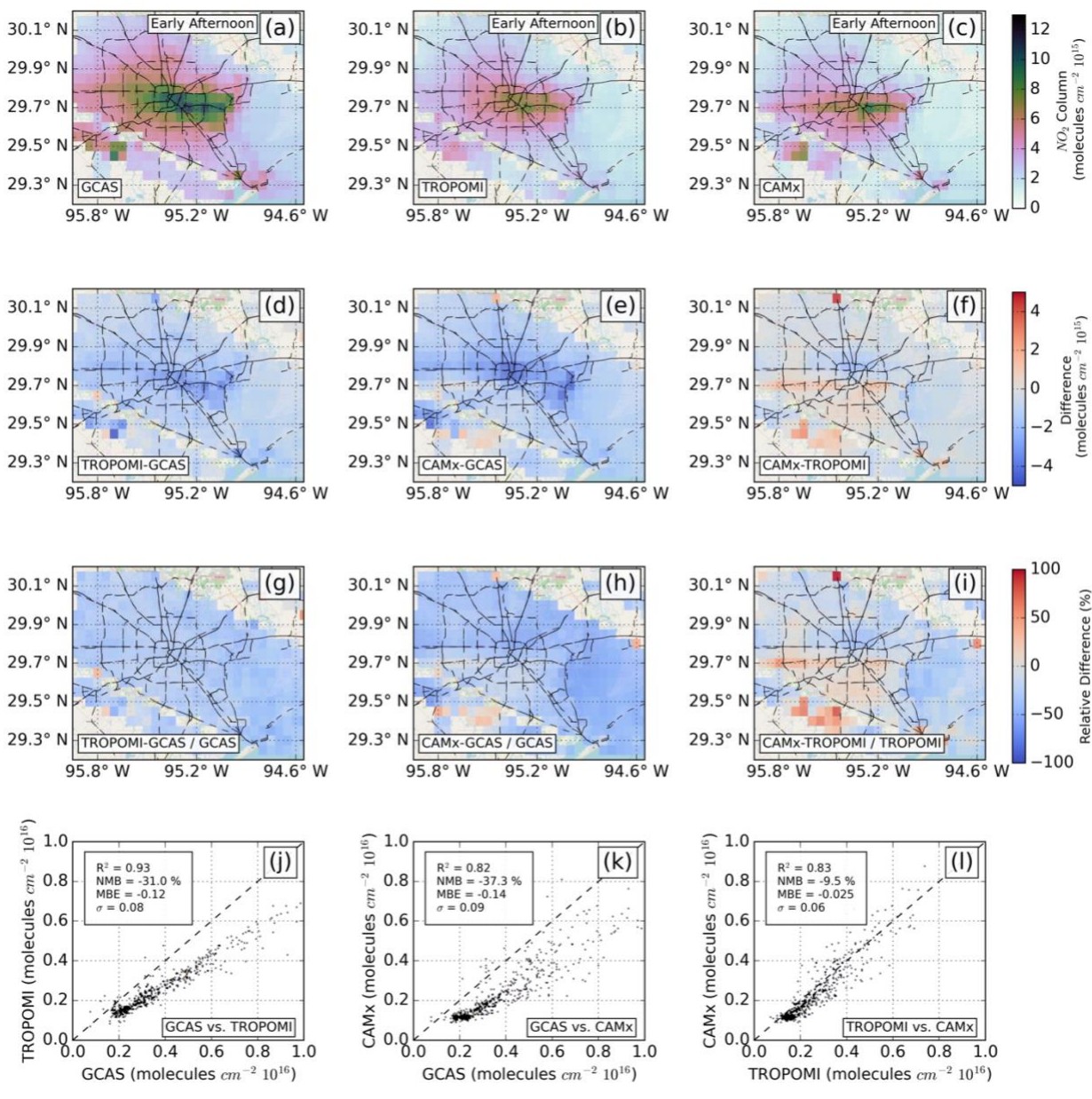

**Figure 6: Spatial distribution of GCAS (A), TROPOMI (B), and CAMx (C) at the 0.05° × 0.05° resolution averaged across the ten GCAS flight days when within 1.5 hours of each TROPOMI overpass representing early afternoon NO₂ columns. Absolute differences between GCAS and TROPOMI (D), GCAS and CAMx**
**(E), and TROPOMI and CAMx (F). Relative differences between GCAS and TROPOMI (G), GCAS and CAMx (E), and TROPOMI and CAMx (F). Scatter density plots of GCAS vs. TROPOMI (G), GCAS vs.**

**CAMx (H), and TROPOMI vs. CAMx (I). © OpenStreetMap contributors *2023*. Distributed under the Open Data Commons Open Database License (ODbL) v1.0.**

Generally, this change in resolution has only a minor effect on the trends discussed in the prior section. Observed $NO_2$ columns from GCAS and TROPOMI have a collocated peak in downtown Houston and $NO_2$ columns from TROPOMI are still systematically biased lower compared to GCAS. Simulated $NO_2$ columns from CAMx are clearly lower than GCAS in the area directly west of downtown and slightly higher southwest of downtown compared to TROPOMI (Fig. 6A-C). Considering the spatial distribution of absolute (Fig. 6D-F) and relative (Fig.

6G-I) differences between the three products, the low bias in TROPOMI compared to GCAS is generally homogenous throughout the domain. On the other hand, there are clear peaks in negative biases in downtown and western Houston when comparing CAMx to GCAS and in some areas southwest of downtown biases are small and positive. Averaging observations to this coarser resolution improved the correlation for all three pairs ($r^2$=0.93, 0.82, and 0.83 for GCAS and TROPOMI, GCAS and CAMx, and TROPOMI and CAMx, respectively) while the biases

remained comparable to what was found in the comparison at a finer resolution (Fig. 5J-L).

**3.5 Weekend vs. weekday patterns across the datasets**

Three of the ten GCAS flight days occurred on weekends (September 11[th], 25[th], and 26[th]) and observations from GCAS and TROPOMI – along with simulated $NO_2$ columns from CAMx –exhibited different patterns on weekends versus on weekdays (September 1[st], 3[rd], 8-10[th], 23[rd], 24[th]). This difference in observed and simulated patterns is

605 explored in greater detail in this section, first through comparisons to Pandora measurements (Fig. 7) and then through spatial comparisons of the products on weekdays versus on weekends (Fig. 8). When interpreting these results, it should be considered that weekend data is limited to only three days. This data sparsity introduces a high degree of uncertainty in conclusions derived from this analysis. Day to day changes in meteorological conditions are likely responsible for some of the exhibited differences so they cannot solely be attributed to differences in emission

patterns.

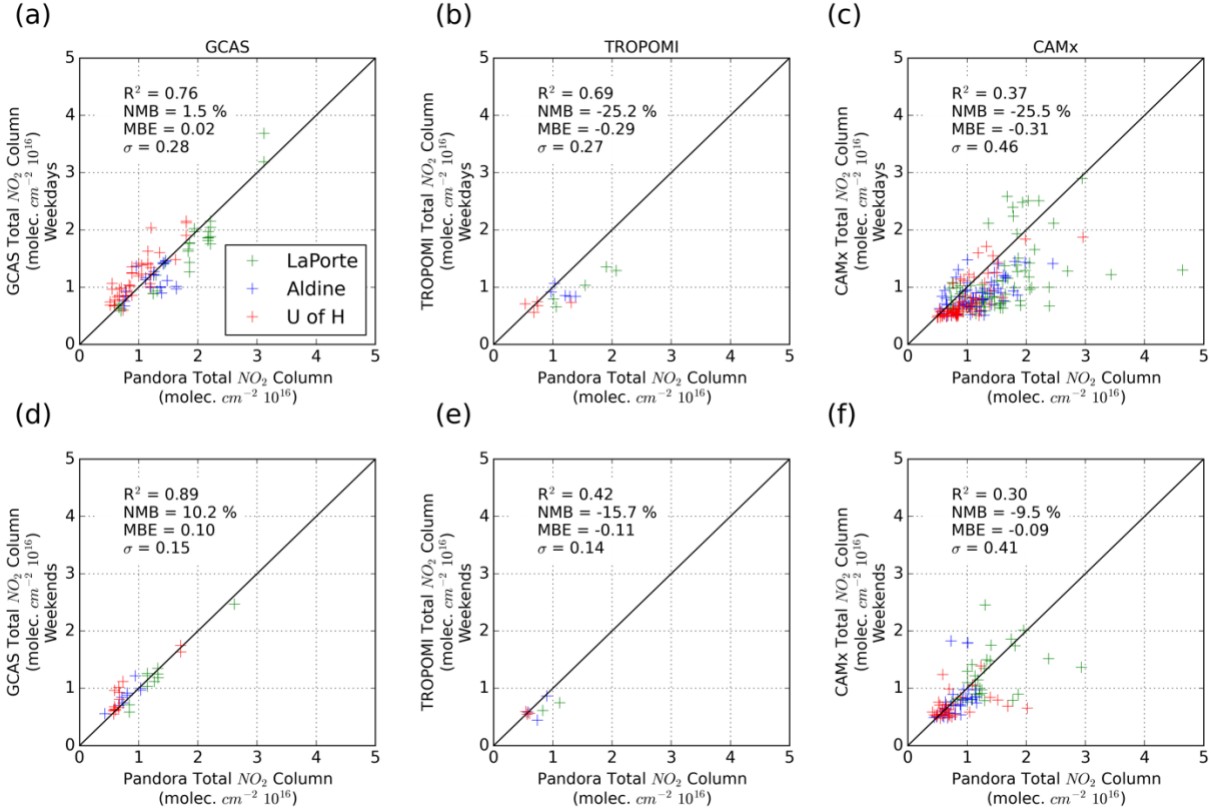

**Figure 7: Comparison of GCAS (A), TROPOMI (B), and CAMx (C) to Pandora on weekdays and of GCAS (D), TROPOMI (E), and CAMx (F) to Pandora on weekends. Data from all possible overpasses coincident within 15 minutes of a Pandora observation are considered. GCAS flight times generally ranged from 8:00 AM-4:00 PM CDT. TROPOMI overpasses occurred around 1:30 PM local time.**

First, we consider how comparisons of the observational datasets – GCAS and TROPOMI – with Pandora change on weekends compared to weekdays. Biases for both GCAS and TROPOMI become more positive on weekends, NMB=10.2% and NMB=-15.7%, respectively, than on weekdays, NMB=1.5% and NMB=-25.2%. GCAS observations are slightly better correlated to Pandora measurements on weekends ($r^2$=0.89 versus $r^2$=0.76); however, TROPOMI observations are worse correlated ($r^2$=0.42 versus $r^2$=0.69) that is likely attributable to a limited number of observations that are at lower $NO_2$ column levels with limited dynamic range. Overall, biases are slightly worse for GCAS and better for TROPOMI on weekends; however, given the small number of measurements it is unclear whether this pattern is attributable to meteorological conditions or if it is attributable to some systematic bias in the instruments.

Simulated $NO_2$ columns from CAMx exhibit clearer weekday versus weekend patterns, and since these simulated columns are available for every hour of the day there is a greater number of measurements to support these findings

than for the two observational datasets. While the correlation is slightly degraded on weekends ($r^2$=0.30 versus $r^2$=0.37) the negative bias in simulated columns compared to Pandora measurements is reduced on weekends (NMB=-25.5% versus NMB=-9.5%).

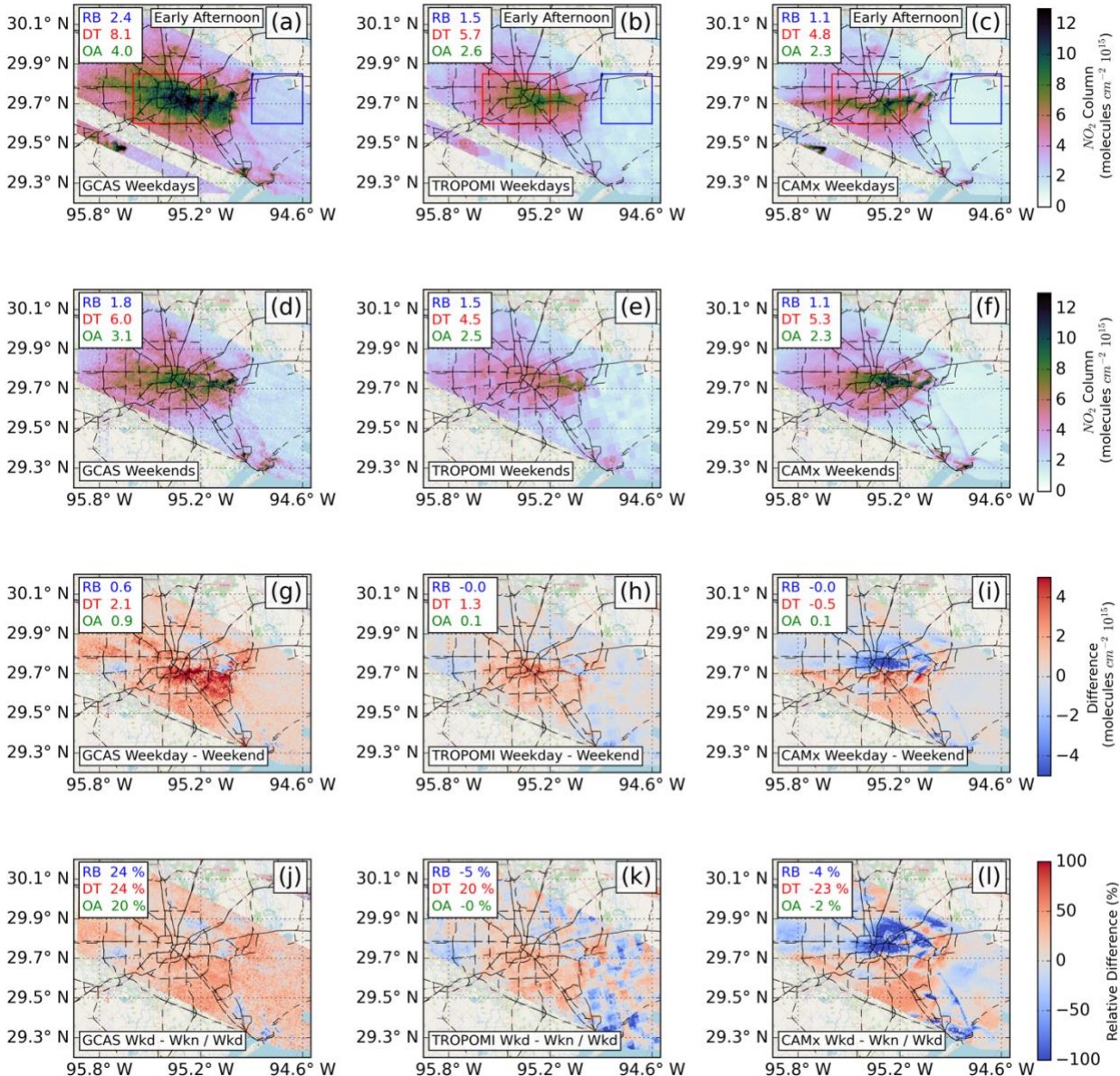

**Figure. 8: Spatial Distribution of GCAS, TROPOMI, and CAMx NO₂ columns on weekdays (A-C), weekends (D-F), and the absolute difference between weekdays and weekends (G-I) and relative difference (J-L). Data are averaged across the GCAS flight days corresponding to weekdays or weekends when within 1.5 hours of each TROPOMI overpass representing early afternoon NO₂ columns. We identify three distinct areas:**

**downtown or "DT" (red), the low emissions East Galveston Bay or "RB" (blue), and all other areas or "OA" (green) and calculate the averages in the top left of each chart. © OpenStreetMap contributors 2023. Distributed under the Open Data Commons Open Database License (ODbL) v1.0.**

GCAS and TROPOMI observations of $NO_2$ column concentrations are higher on weekdays (Fig. 8A, B) than on weekends (Fig. 8D, E). This is true in downtown Houston and the rural bay where weekday GCAS observations are 2.1 molecules $cm^{-2}$ $10^{15}$ (24%) and 0.6 molecules $cm^{-2}$ $10^{15}$ (24%) higher, respectively, on weekdays than on weekends. In other areas of Houston, GCAS observations on weekdays are higher than weekends but not to the same degree (20%). A similar pattern occurs for TROPOMI, in downtown Houston TROPOMI columns are 20% higher on weekdays than on weekends but comparable in other areas and -5% lower in the rural bay. This comparison again implicates some underestimated weekday source of $NO_2$ in CAMx that is of great importance in the western area of Houston; however, due to the lack of data on weekends – that is apparent in the discontinuities in the weekend $NO_2$ column concentrations of TROPOMI – it is difficult to examine this quantitatively.

Comparing weekday columns simulated from CAMx with weekend columns, we find that the mean concentrations for the three defined areas are nearly identical (Fig. 8C and F), although columns on weekdays are higher south and southwest of downtown while columns on weekends are higher within downtown. These spatial patterns are further revealed in the difference plots (Fig. 8I and 8L) where the difference in weekday versus weekend values appear to be split right along I-10; north of I-10 weekday values are much lower than weekend values while south of I-10 the opposite is true. This difference is likely attributable to different meteorological conditions on these days. Overall, simulated CAMx columns are substantially lower than GCAS and TROPOMI on weekdays but more similar on weekends implying that weekday emissions may be underestimated in the TCEQ inventory.

**3.6 Relevance to TEMPO: Diurnal patterns in column and surface NO₂**

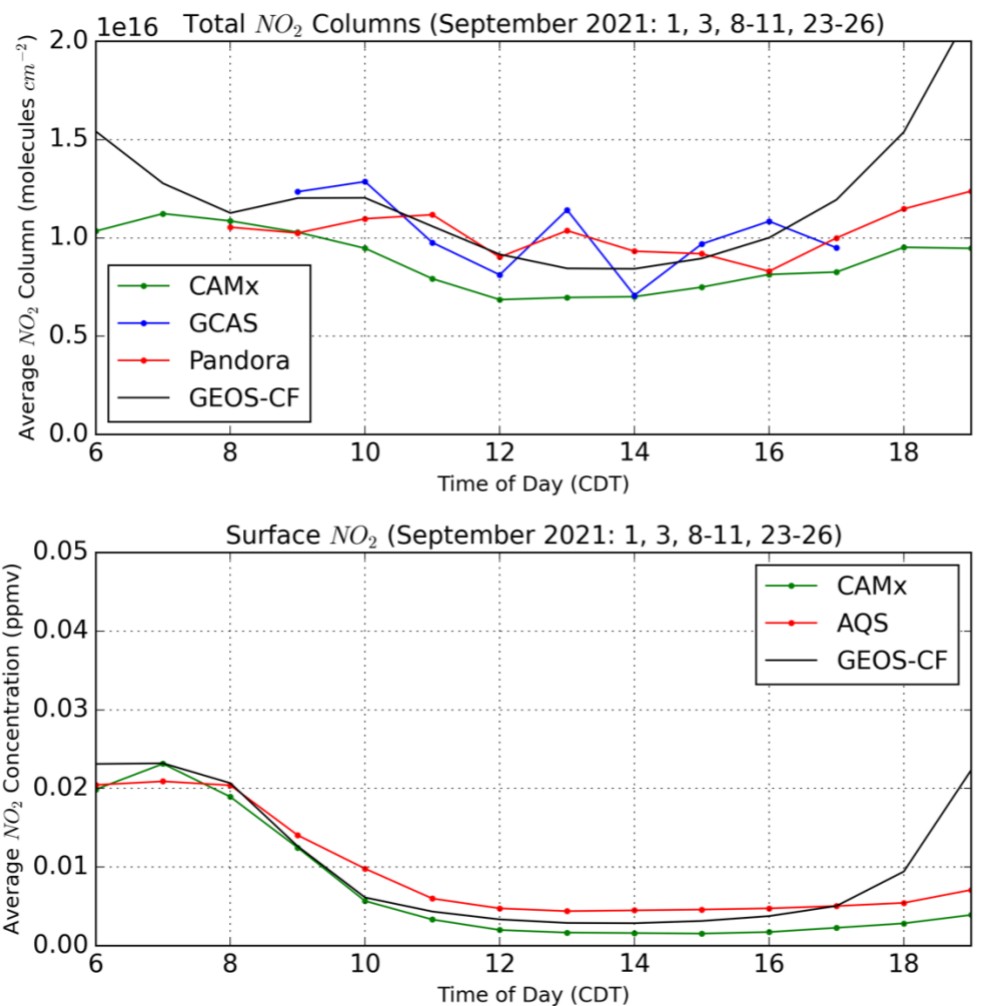

**Figure 9: Diurnal patterns in Total NO₂ columns (top) averaged across the three Pandora sites and 10 flight days from CAMx (green), GCAS (blue), Pandora (red), and GEOS-CF (black). Diurnal patterns in surface-level NO₂ concentrations (bottom) in downtown Houston for CAMx and GEOS-CF averaged across the 10 flight days and across all monitors in Harris County for AQS surface-level monitors (red).**

Lastly, we characterize the diurnal profiles of simulated and observed NO₂ columns during the TRACER-AQ campaign in downtown Houston (Fig. 9). First, considering column concentrations, we find generally good agreement during the early morning (8:00 -10:00 local time) across the two simulated datasets (CAMx and GEOS-CF) and two observational datasets (GCAS and Pandora). Interestingly, between 9:00 and 11:00 local time, Pandora column measurements show a slight increase, while model simulations show a slight decrease during the same time

interval. During midday and the afternoon (11:00 – 16:00 local time) – that corresponds to the period with the most GCAS observations – GEOS-CF columns generally agree well with Pandora observations. In the evening (17:00 – 19:00 local time), GEOS-CF columns have a substantial high bias across these flight days. The GEOS-CF mismatch in the evening has implications for TEMPO $NO_2$ evening retrievals if this is a persistent bias in other urban areas since satellite instruments are especially sensitive to *a priori* assumptions at low sun angles.


Second, considering surface concentrations, we see a similar trend. Generally, there is great agreement across the three datasets (CAMx, AQS observations, and GEOS-CF) in the early morning (6:00 – 9:00 local time) before they begin to diverge with the two simulated produces maintaining comparable magnitudes with low biases compared to surface monitors. At around midday to the afternoon (12:00 – 17:00 local time) both simulated products have a low

bias compared to observed surface-level $NO_2$; however, the bias in CAMx concentrations is worse. Some of the apparent low bias may be related to an artificial high bias in $NO_2$ chemiluminescence surface monitors (Dunlea et al., 2007; Lamsal et al., 2008). In the evening (18:00 – 19:00 local time), surface-level $NO_2$ from GEOS-CF climbs rapidly; however, observed $NO_2$ from the AQS and simulated $NO_2$ from CAMx increase only slightly. The large increase in $NO_2$ in GEOS-CF in the evening appears at both the surface and in the column potentially indicating

issues capturing boundary layer dynamics.

**4 Conclusions**

This study leveraged observational datasets of $NO_2$ column densities from three instruments – Pandora ground-based spectrometers, the airborne GCAS instrument, and the satellite TROPOMI instrument. These instruments were used to investigate $NO_2$ column densities in Houston, TX during the September 2021 TRACER-AQ campaign

and to characterize strengths/weaknesses and uncertainties in the respective datasets. These observational datasets were then compared to simulated $NO_2$ columns from CAMx to characterize the performance of the simulation and to identify potential under- or overestimates of emissions in the simulation. We find that GCAS has strong agreement with Pandora instruments ($r^2$=0.79 and NMB=3.4%) during its overpasses and that TROPOMI also has strong performance but an important low bias – consistent with validation by the European Space Agency (Verhoelst et al.,

2021) – across the urban Houston locations ($r^2$=0.73 and NMB=-22.8%). This low bias in TROPOMI observations persists despite the inclusion of an air mass factor derived from the CAMx simulation. When comparing different versions of TROPOMI we find differences between the v2.3.1, v2.4.0, and NASA MINDS product and find that the MINDS ($r^2$=0.69 and NMB=-18.2%) and version 2.3.1 ($r^2$=0.72 and NMB=-18.3%) products – with the CAMx AMF – performs comparably and both outperform version 2.4.0 considering bias albeit with slightly worse

correlation. The performance of the CAMx simulation varied depending on the day, but overall, simulated $NO_2$ columns were more poorly correlated and more negatively biased compared to Pandora measurements than the observational datasets ($r^2$=0.34 and NMB= -21.2%). Notably, this low bias in CAMx simulated $NO_2$ columns improved on weekends (NMB=-9.5%) – albeit over a limited number of days. This improvement on weekends implicates that a source that emits in greater amounts on weekdays (e.g., heavy-duty vehicles) could be

underestimated in the TCEQ inventory; however, we cannot say this conclusively given the limited number of observations on weekends. The poor correlation in the simulated $NO_2$ columns is likely attributable to minor wind directional errors – simulated wind direction had a MAE that ranged from 14° to 32° when compared to observations – and spatial correlations over larger extents match well.

When we compare the spatial distribution of TROPOMI observations to GCAS (Fig. 3 and 4) we find that the low bias in TROPOMI $NO_2$ columns is perhaps stronger than the low bias implied at the three Pandora sites – this could be a resolution constraint of the coarser TROPOMI product that is unable to capture the fine-scale features in $NO_2$ column concentrations that GCAS is able to. If coarse resolution is responsible for this low-bias, new instruments on geostationary satellites from missions like the NASA TEMPO mission could be leveraged to further improve

satellite-derived estimates of urban $NO_2$ in cities like Houston. CAMx comparisons to GCAS, when extended beyond the limited number of Pandora sites, indicate that the CAMx simulated low bias could be substantially worse than at the Pandora sites (-32%) in downtown and west of downtown Houston. This overall underestimate in the CAMx simulations is potentially attributable to a number of confounding factors including an inability of the WRF simulation to capture local meteorology – WRF simulated wind speeds had only modest correlation with

observations ($R^2$=0.26) although there was little systematic bias (MBE = -0.02) – and an underestimate of emissions in sectors that are more spatially located in downtown and western Houston like on-road mobile emissions. We also consider differences in the diurnal profiles of surface and column $NO_2$ across multiple datasets and find that the performance of CAMx is at its worst in the late morning and early afternoon and performance is better during other times of the day.

There is a clear negative bias in the CAMx simulated $NO_2$ columns compared to GCAS observations. Although we primarily evaluate the performance of WRF meteorology at the surface, we also briefly investigate model vertical structure for five ozonesondes from different locations and days (Fig. S14-S18) and find great agreement in temperature and pressure; however, there is more mixed agreement in the ozone mixing ratio, wind speed, and wind direction. Future evaluation of 3D model simulated vertical structure for $NO_2$ using observations from NASA – such

as measurements from the High Spectral Resolution Lidar 2 (HSRL-2) instrument, the Tropospheric Ozone Lidar Network (TolNet), or TRACER-AQ – may be helpful for diagnosing the distinct influences of emissions, meteorology, and chemistry on column $NO_2$. A previous study by Liu et al., (2023) has investigated this for the TRACER-AQ campaign in Houston, albeit with a different chemical transport model, and found generally good agreement in potential temperature but an underestimate of ozone in the free troposphere. We note that the YSU scheme used in the

WRF-CAMx simulation (Table S1) has been shown to underestimate PBL height in the Houston area during the TRACER-AQ campaign (Liu et al., 2023) which would likely impact the vertical distribution of $NO_2$. Given the worse performance of WRF-CAMx at the surface (NMB=-59%) than for the columns (-22%), if the vertical mixing scheme has poor performance we suspect it to be due to overmixing leading to the rapid removal of surface-level $NO_2$. Additionally, the low bias in the TROPOMI observations compared to Pandora and GCAS merits further investigation;

the role of algorithm and resolution could be considered by comparing different versions and finer-resolution

geostationary observations in the future beyond what is considered in this study. The reference background $NO_2$ from TROPOMI used in GCAS could also introduce error into these results that should be considered. Given the fine resolution of GCAS observations and CAMx simulated column concentrations there is potential for investigations into how air pollution is inequitably distributed across different populations in Houston and how specific sources contribute to these inequities. The findings presented here imply that TROPOMI derived $NO_2$ column concentrations may be underestimated in Houston if not corrected for in applications such as exposure assessments, and $NO_X$ emissions derivations.

This analysis benefitted from three independent measurement datasets (i.e., Pandora, TROPOMI, and GCAS) that were critical to isolate the negative biases in TROPOMI and CAMx although we note that negative biases in TROPOMI have been mentioned in earlier literature (e.g., Verhoelst et al, 2021) and in the quarterly issued operational validation reports (available at https://mpc-vdaf.tropomi.eu/). It is common to consider TROPOMI measurements as accurate representation of $NO_2$ column concentrations; however, if we had done so in this study, we would have failed to identify the substantial negative bias in the CAMx simulation of column concentrations. Observations from multiple Pandora instruments and GCAS overpasses made it possible to isolate negative biases in TROPOMI and CAMx. While there are some errors in the meteorology – notably only a modest correlation between simulated and observed wind speed, albeit with little systematic bias, and mixed capturing of vertical structure compared to ozonesondes observations – these errors are unlikely to fully explain the low bias in simulated $NO_2$. Given the relatively minimal biases in WRF simulated wind speed and direction at the surface compared to observations, low $NO_2$ biases in the simulated CAMx column concentrations imply that current TCEQ $NO_X$ emissions inventories in the Houston area used to drive the CAMx simulation may be underestimated, and that this underestimation is likely attributable to a source with weekday-weekend differences and correlated with roadways and/or population density.

**Acknowledgements**

The authors of this paper acknowledge Elena Lind, Alex Kotsakis, the NASA Pandora Project and LuftBlick for their contribution in deploying, operating, and processing data from the Pandora spectrometers during TRACER-AQ, the NASA Tropospheric Composition Program and the Texas Commission on Environmental Quality for TRACER-AQ support, and the TRACER-AQ science team for their useful contributions. We also acknowledge the use of Google Earth for the background map used in Fig. 1. The authors acknowledge the use of OpenStreetMaps for the background maps in Fig. 2, 4, 5, 6, 8 and S2-14. The authors also acknowledge funding from the NASA Atmospheric Composition Modelling and Analysis Program (ACMAP) (80NSSC23K1002). The preparation of this manuscript was funded by a grant from the Texas Air Quality Research Program (AQRP) at The University of Texas at Austin through the Texas Emission Reduction Program (TERP) and the Texas Commission on Environmental Quality (TCEQ). The findings, opinions and conclusions are the work of the authors and do not necessarily represent findings, opinions, or

conclusions of the AQRP or the TCEQ. This work contains modified Copernicus Sentinel-5 Precursor data processed by KNMI and post-processed by George Washington University.

**Code Availability**

The scripts used to process these data for the intercomparison are available by correspondence with the authors upon reasonable request.

**Data Availability**

Ground level EPA AQS observations are available from EPA Air Data: https://www.epa.gov/outdoor-air-quality-data. GEOS-CF data are available from the GrADS data server: https://opendap.nccs.nasa.gov/dods/. TROPOMI $NO_2$ column data are publicly available from the Copernicus Data Space Ecosystem: https://dataspace.copernicus.eu. GCAS and CAMx data are publicly available via NASA's Atmospheric Sciences Data Center: https://doi.org/10.5067/ASDC/SUBORBITAL/TRACERAQ/DATA001/GV/AircraftRemoteSensing/GCAS_1 as are Pandora data https://asdc.larc.nasa.gov/project/TRACER-AQ/TRACERAQ_Pandora_Data_1.

**Author Contributions**

D.G., developed the project design, J.J. and G.Y. set-up and conducted the WRF-CAMx simulations, L.J. and the TRACER-AQ science team measured and processed the GCAS and Pandora Data, D.G downloaded and processed the TROPOMI Data and regridded all data to the WRF-CAMx grid, M.O.N further processed the data to match temporally and spatially, conducted the intercomparison, wrote the manuscript, and generated the figures except Fig. 1 and Fig. S1 that were generated by J.J. and D.G., respectively, B.D. gave feedback on the methodology, all authors edited the manuscript and gave feedback on the figures.

**Competing Interests**

The authors declare that they have no conflict of interest.

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
