# Peer review of "An intercomparison of satellite, airborne, and ground-level observations with WRF-CAMx simulations of $NO_2$ columns over Houston, TX during the September 2021 TRACER-AQ campaign"

_EGUsphere, 2023_

## Referee Comment (RC2)

**General Comments**

The authors compare WRF-CAMx simulated column $NO_2$ at 444 m resolution with Pandora, GCAS, and TROPOMI observations. The model is biased low against Pandora and has poor agreement with GCAS $NO_2$ in downtown Houston, suggesting a potential underestimate of vehicle $NO_x$. The main general comment is that the authors should provide more support for their model performance before making conclusions about emissions. Could the authors compare the model vertical profiles of temperature and relative humidity against weather soundings, and/or the model ozone profiles against TRACER-AQ ozonesondes? This would better support that the difference between GCAS and the model is due to emissions. In addition, it would be helpful to show the model comparison against any surface meteorology or air quality data ($NO_2$, ozone?) available during the campaign. With this major revision and the comments below, this paper provides a useful study of the pitfalls of using TROPOMI for constraining emissions and the benefit of high resolution modeling and aircraft data.

**Specific Comments**

Line 43 – Does $NO_2$ exceed the NAAQS in Houston? Or do you mean that elevated $NO_2$ is indicative of higher emissions from combustion sources generally and is a precursor for ozone and $PM_{2.5}$? Where does Houston fall in compliance for the NAAQS for these species?

Line 84 – What do you mean by "remote-sensing columns"? Satellite?

Line 159 – This is confusing. Why would you compare Pandora to re-gridded GCAS and not the native resolution GCAS?

Line 160 – 30 minutes seems like a long time given that direct sun Pandora data are more frequent and could be matched more closely. What is the variability in the data over that window and is it really reasonable to use such a long time period?

Line 190 – For comparison to TROPOMI, you need to regrid the model to the coarser TROPOMI resolution of 3.5x5.5 $km^2$, otherwise the comparison will certainly look poor.

Line 242 – Can you explain what you mean by "Third, we reprocess link-based on-road mobile emissions for the higher resolution domains."? What is the native resolution of the emissions? Is this some sort of down-scaling procedure?

Line 249 – High degree of uncertainty does not seem like a good reason to remove fire or lightning emissions. Does the model suggest that they are unimportant in Houston and thus you remove them to save computational expense?

Line 256 – Why are you including GEOS-CF in this comparison? You haven't given us a motivation to do this.

Line 332 – Do you think the model successfully captures horizontal advection and vertical mixing in Houston? How could errors in transport relate to the model underestimate in NO2 columns? Does the model perform significantly differently on days with slower or faster wind speeds?

Line 348 – Again, any windspeed or direction dependence? What about hour of day dependence?

---

## Author Comment (AC1)

**Response to the reviewers of:**

*An intercomparison of satellite, airborne, and ground-level observations with WRF-CAMx simulations of NO₂ columns over Houston, TX during the September 2021 TRACER-AQ campaign*

**Note to all:**

We thank the reviewers of the manuscript titled above for their detailed and helpful reviews. In this document we respond to their comments using the following notation: plain text indicates the reviewers' comments, **bolded text** indicates our responses, and *italicized text* indicates quotations from the original and updated manuscripts. For quotations, we will indicate whether they come from the original manuscript or the updated manuscript using "before" and "after", respectively; new text is indicated with red coloring. We have endeavoured to address the concerns of the reviewers and believe that these changes have improved the quality of the work.

During our revisions, we made three further unsolicited improvements. First, we had originally presented results from a version of the GCAS dataset with a reference value based on a prior version of TROPOMI algorithm; this change minimally impacted our results and does not affect our high-level conclusions, but we did want to draw attention to it to explain why the numbers slightly changed in section 3. Second, when comparing GCAS values to Pandora we added a stratospheric vertical column $NO_2$ component based on TROPOMI; however, this did not account for the column in the tropospheric amount above the aircraft. We thus add an additional partial column $NO_2$ above the aircraft (~9 km) and below the tropopause (~18 km) based on the CAMx simulation. This partial column is $0.57 \times 10^{15}$ molecules cm$^{-2}$ so it has a minimal impact on our GCAS and CAMx Pandora comparisons in an urban area such as Houston which typically has tropospheric vertical column $NO_2$ values exceeding $5 \times 10^{15}$, and does not affect our final conclusions. Lastly, we apply a diurnal variation to the stratospheric vertical column derived from work by K.-F. Li et al. (2021) to account for diurnal patterns in stratospheric $NO_2$ that is equivalent to $1.34 \times 10^{14}$ molecules cm$^{-2} \times dt$ where $dt$ is the difference between the time in hours – in either GCAS or CAMx – and 13:30 – the approximate TROPOMI overpass time. Applying this diurnal variation to the constant stratospheric vertical column from TROPOMI better captures lows in stratospheric $NO_2$ earlier in the day and higher values in the afternoon and evening.

**Anonymous Referee #1:**

**Major Comments:**

[1] My major concern is that WRF-CAMx is performed at a very high resolution (444 x 444 m$^2$) but is lack of enough meteorology evaluations, which is very important for column comparisons. In this case, it's difficult to attribute the model bias to emission inventory, and arbitrary to conclude the potential underestimation of vehicular NOx emissions from the Texas state inventory.

**We thank anonymous referee #1 (RC1) for their comments. We agree that the conclusions made in the original manuscript would be better supported if meteorological conditions in the WRF simulation were evaluated compared to observations. We have now evaluated the WRF meteorology and find excellent agreement. To address this comment, we have now added four tables and ten figures to the supplement (Table S5-S8 and Fig. S2-S11) that compare WRF simulated wind speed, wind direction, temperature, and water vapor mixing ratio to observations at sixteen ground-level monitors around Houston. We did not find any significant biases in the WRF simulation. For all days, the WRF wind direction mean bias error (MBE) was -8°, wind speed MBE was -0.02 m/s, temperature MBE was 0.39 K, and water vapor mixing ratio MBE was -1.45 g/kg. Across non-cloudy days, daily WRF wind direction mean absolute error (MAE) ranged from 14° to 29°, wind speed MAE ranged from 0.94 m/s to 1.35 m/s, temperature MAE ranged from 0.93 K to 1.18 K, and water vapor mixing ratio MAE ranged from 0.87 g/kg to 3.11 g/kg. We have also included these tables and figures at the end of this document for convenience.**

**We find excellent agreement between WRF and observations at the surface on non-cloudy days (on cloudy days performance is poorer but data from these days are mostly excluded because GCAS and TROPOMI were unable to observe NO₂ columns in most pixels). Specific quantitative performance metrics are included in the new text added below [A].**

**To address your concerns, we have made the following changes to the text: [A]We have indicated that we have done this comparison in the methods section and [B] qualified our results and conclusions with it throughout the text. [C] Lastly, we have softened the conclusion regarding the potential underestimation of vehicular NOₓ emissions in the abstract and added explanation for why mobile sources are named specifically.**

[A]

[revised manuscript text omitted]

[2] Also, authors should clarify if the $NO_2$ columns are tropospheric or the total columns, to avoid misunderstanding.

**Thank you for this suggestion. We agree with you that we can be clearer in distinguishing when we are presenting results for tropospheric $NO_2$ columns compared to total columns. To address this concern, we have added some new text at the beginning of section 3.1 [A]. Additionally, we have updated Figure 2, 3, 7, and 9 – the comparisons to Pandora – to explicitly mention that we are comparing total $NO_2$ columns. These updated figures have been included at the end of this document for convenience.**

[A]

**New Text:**
*When comparing the observational and simulated datasets to Pandora observations we consider the total column $NO_2$ – we add a stratospheric component from TROPOMI to the tropospheric column $NO_2$ of GCAS and CAMx to total column as discussed in the methodology. For TROPOMI, we use an AMF derived from the CAMx simulation to calculate a tropospheric $NO_2$ column from TROPOMI; following the TROPOMI users guide, we multiply the total averaging kernel by the ratio of the total air mass factor to the tropospheric air mass factor. We difference the total column $NO_2$ from TROPOMI with the tropospheric column to estimate a constant stratospheric $NO_2$ column amount that we apply to GCAS and CAMx when we compare them to Pandora; this corresponds to a mean value of $3.0 \times 10^{15}$. For GCAS, we apply an additional amount to account for the $NO_2$ column in the upper troposphere – above the aircraft and below the tropopause. We calculate the column of the 3 highest levels of CAMx that correspond to 9400-18100 m above sea-level – that is roughly the same height that GCAS flies at of around 9100 m – and apply this to the GCAS results; this corresponds to a value of $0.57 \times 10^{15}$ molecules $cm^{-2}$. For all results that include comparison to Pandora we present total $NO_2$*

*columns; for results where we only intercompare GCAS, TROPOMI, and CAMx we compare the tropospheric column. All statistical measures (e.g., $R^2$) are defined in the supplement.*

**Minor Comments:**

[3] Line 21: please give the full name of TROPOMI in the first instance.

**Replaced as suggested.**

[4] Line 27 – 29: I doubt this conclusion because there are no meteorology evaluations of the model.

**We understand your concern that this conclusion was weak as originally presented due to a lack of meteorological evaluation of the model. To address this concern, we have compared WRF-simulated meteorology to ground-level observations and softened the language used to make this conclusion. Detailed updates to the text to address this concern are presented in response to your major comments [1].**

[5] Line 102: please specify if the column is tropospheric, or the total.

**Thank you for this comment, we compare total column amounts. This has been clarified in the text as presented in response to your major comment [2].**

[6] Line 113 – 114: not sure what this sentence means. Please rephrase it.

**Sorry for this confusion. We were trying to emphasize that two sites had multiple instruments, and if we averaged all observations together, we would be unevenly weighting the values at certain measurement sites. We have now updated the text to clarify this [A].**

[A]

**Before:**
*The study was designed to have two Pandoras operating at once during the study. This avoids unevenly weighting coincidences that would arise from intercomparisons with airborne, TROPOMI and CAMx datasets.*

**After:**
*The study was designed to have two Pandoras operating coincidently at each site during the campaign; however, due to instrument failures an uneven number of observations were obtained at each site. In order to evenly weigh the observations between the three sites, we select data from a single Pandora instrument at each site.*

[7] Line 164 – 166: in TROPOMI, the separation of tropospheric and stratospheric column is also another big source of uncertainty. Using the stratospheric NO2 columns derived from TROPOMI can bring in unpredicted bias as well. Can you give analyses on the stratospheric NO2 columns here? What's the data range and spatial distributions?

**We agree with you that the separation of tropospheric and stratospheric columns in TROPOMI could be a source of uncertainty and that using stratospheric NO₂ columns derived from TROPOMI can introduce bias as well. To address this concern, we quantify the data range of stratospheric NO₂ by reviewing past studies and citing results summarized Geddes et al. (2018). Additionally, we include the Boersma et al. (2018) suggestion of the structural uncertainty in the stratospheric estimate ($0.5 \times 10^{15}$ molecules cm$^{-2}$) so that readers can keep this uncertainty in mind when reviewing our results. We have added new text in the methodology section to address this comment [A]:**

[A]

**New Text:**
*This stratospheric vertical column NO₂ amount of $3.0 \times 10^{15}$ is typical for Houston during summer (Geddes et al. 2018). Boersma et al. (2018) suggests that $0.5 \times 10^{15}$ molecules cm$^{-2}$ is the upper limit of structural uncertainty in the stratospheric estimate; this uncertainty should be considered when reviewing results that compare total column amounts (i.e., results comparing GCAS and CAMx to Pandora).*

[8] Sect. 2.4: at such high spatial resolution, how does meteorology work?

**We agree that the description of the fine resolution WRF meteorological simulation could be explained with more detail. The important consideration in conducting WRF simulations at a fine spatial resolution is the configuration of physical schemes that are sensitive to grid spacing, i.e., convective cumulus parameterization. We turned off the cumulus parameterization scheme for the 4 km, 1.33 km and 444 m grids because WRF physics can explicitly simulate convection for these grids. We turned on the cumulus parameterization for grids coarser than 4 km to account for sub grid-scale convection. We kept other physics options consistent across all grids; these physics options are included in Table 1 of the supplement and we have also included it in this document for convenience. We have added new text to section 2.4 in response to this comment [A].**

[A]

**New Text:**
*Conducting WRF simulations at fine spatial resolutions (i.e., 4 km, 1.333 km, and 0.444 km) requires careful consideration of physical schemes that are sensitive to grid spacing. We turn off the convective cumulus parametrization scheme for the fine grids because WRF can explicitly simulate convection for them. For coarser grids we turn on the cumulus parametrization to account for sub grid-scale convection. The other physics options (Table S1) are kept consistent across the different resolutions.*

[9] Line 248 – 250: what's the resolution of the emission inventory? For CEMS, they are point sources with exact geophysical locations. How about other sources? The in-compatible spatial resolution between model and inventory can be a big issue.

We agree that spatial downscaling emissions inventories is an important consideration. In this project, we were careful in developing a gridded emissions dataset. Specifically, beyond Point sources that are mapped based on their geophysical location. Line source sectors – on-road vehicle and shipping emissions – are mapped based on known roadway networks, ship tracks, and traffic patterns. Area sources such airports, railyards, and biogenic emissions are mapped to their known locations are re-gridded to the 444 m resolution. Other sectors such as (3) area, oil and gas, and off-road vehicle emissions are retained at the 4 km grid resolution provided to us by TCEQ. We have added new text to section 2.4 that further clarifies how we regrid the different sectoral sources beyond what we detailed in the original manuscript [A] and we have updated Table 3 to explicitly define the resolutions of the inventory; we include Table 3 at the end of this document for convenience.

[A]

**New Text:**

*Specifically, all point sources are geo-located to the grid cell containing the source. On-road mobile source emissions and shipping emissions were provided for individual links which we allocated to 444 m grid cells, and are based on known roadway networks, ship tracks, and traffic patterns. Airport and railyard emissions were allocated to 444 m grid cells within the property boundary. Other sources retained the 4 km grid resolution provided by the TCEQ.*

[10] Line 277 – 279: TROPOMI and GCAS AMF calculation not only include geometry, but also on *a priori* information from models. I'm just curious how does Pandora calculate the air mass factor without the *a priori* NO2 vertical profiles? Can you elaborate more on this?

**Pandora AMFs are not reliant on an a priori profile as the data we are using is only in direct-sun mode in cloud-free scenes. AMF for Pandora is analogous to pathlength through the atmosphere relative to the vertical path. Since all the signal is from direct sun path (extremely minimal scattering), this is purely geometric. The a priori profiles are needed for the scattered light DOAS (TROPOMI, GCAS) because the instruments are not equally sensitive throughout the atmospheric column so the AMF accounts for how sensitive we are to each layer of the atmosphere (dependent on factors like viewing/solar geometry and surface reflectivity) and which fraction of the trace gas is expected from this layer (the a priori profile). We have added some new text to clarify this in the manuscript:**

**New Text:**

*Pandora AMFs are not reliant on an a priori profile as the data we are using is only in direct-sun mode in cloud-free scenes and since all the signal is from direct sun path – that has extremely minimal scattering – this calculation is purely geometric.*

[11] Line 613 – 614: This, further emphasize the needs of meteorology evaluations in the model.

**We agree that this statement would be better supported with evaluations of the simulated meteorology. We have conducted these and discuss detailed results in response to your**

**major comments [1]. We have additionally updated this statement to reflect these evaluations.**

**Before:**
*The poor correlation in the simulated NO$_2$ columns is likely attributable to minor wind directional errors and spatial correlations over larger extents match well.*

**After:**
*The poor correlation in the simulated NO$_2$ columns is likely attributable to minor wind directional errors – simulated wind direction had a MAE that ranged from 14° to 32° when compared to observations – and spatial correlations over larger extents match well.*

**Anonymous Referee #2:**

**Major Comments:**

[1] The main general comment is that the authors should provide more support for their model performance before making conclusions about emissions. Could the authors compare the model vertical profiles of temperature and relative humidity against weather soundings, and/or the model ozone profiles against TRACER-AQ ozonesondes? This would better support that the difference between GCAS and the model is due to emissions. In addition, it would be helpful to show the model comparison against any surface meteorology or air quality data (NO2, ozone?) available during the campaign. With this major revision and the comments below, this paper provides a useful study of the pitfalls of using TROPOMI for constraining emissions and the benefit of high-resolution modeling and aircraft data.

**We thank anonymous referee #2 (RC2) for their helpful and detailed comments and agree that the conclusions about emissions would be better supported if meteorology in WRF was evaluated compared to observations. We have now evaluated the WRF meteorology and find excellent agreement. We have conducted extensive evaluations of meteorology and air quality data at the surface-level to better support conclusions derived from the WRF-CAMx simulation.**

**We have evaluated the meteorological performance of WRF and added four tables and ten figures to the supplement (Table S5-S8 and Fig. S2-S11) that compare WRF simulated wind speed, wind direction, temperature, and water vapor mixing ratio to observations at sixteen ground-level monitors around Houston. We did not find any significant biases in the WRF simulation. For all days, the WRF wind direction mean bias error (MBE) was -8°, wind speed MBE was -0.02 m/s, temperature MBE was 0.39 K, and water vapor mixing ratio MBE was -1.45 g/kg. Across non-cloudy days, daily WRF wind direction mean absolute error (MAE) ranged from 14° to 29°, wind speed MAE ranged from 0.94 m/s to 1.35 m/s, temperature MAE ranged from 0.93 K to 1.18 K, and water vapor mixing ratio MAE ranged from 0.87 g/kg to 3.11 g/kg. We have also included these tables and figures at the end of this document for convenience. We find generally good agreement between simulated and observed meteorology including wind speed, direction, temperature, and water vapor mixing ratio. Additionally, we have softened the conclusion regarding the potential underestimation of vehicular NO$_X$ emissions in the abstract and added explanation for why mobile sources are named specifically. Specific changes to the text regarding this meteorology evaluation are given in response to the major comments [1] of RC1.**

**In response to this comment, we have also conducted an evaluation of simulated NO$_2$ and O$_3$ concentrations at the surface-level. Overall, we found a strong negative bias in simulated hourly surface-level NO$_2$ (NMB=-59%) but good performance in MDA8 O$_3$ (NMB=-2.5%). We have added new text to summarize these results [A] and included them in the main text so that readers can consider this when interpreting the rest of our results and conclusions. Additionally, we have added two new figures (Fig. S13 and Fig. S14) that show these comparisons that we also include at the end of this document for convenience.**

**Comparisons between the vertical profiles of temperature and relative humidity against weather soundings would be interesting, but we believe that it is beyond the scope of this project. Similarly, we believe that comparisons between model ozone profiles and TRACER-AQ ozonesondes are beyond the scope of this project. However, there is ongoing work being done our collaborator, Dr. John Sullivan, who will examine this.**

[A]

**New Text:**
*Additionally, we compare simulated hourly $NO_2$ (Fig. S12) and maximum daily eight-hour average or "MDA8" $O_3$ (Fig. S13) to observations from seventeen TCEQ continuous air monitoring stations (CAMS) operating in Houston. We find poor performance and a strong negative bias in the simulated surface-level $NO_2$ (NMB=-59%) and generally good performance in the simulated surface-level MDA8 $O_3$ (NMB=-2.5%) compared to observations.*

[2] Line 43 – Does NO2 exceed the NAAQS in Houston? Or do you mean that elevated NO2 is indicative of higher emissions from combustion sources generally and is a precursor for ozone and PM2.5? Where does Houston fall in compliance for the NAAQS for these species?

**We agree that the original text was not clear. $NO_2$ itself does not exceed the NAAQS in Houston; however, Houston is in moderate nonattainment of the 8-Hour Ozone (2015) NAAQS and in compliance of the PM2.5 (2012) NAAQS. We meant to indicate that higher levels of $NO_2$ can lead to increased formation of $O_3$ and $PM_{2.5}$ given that it is a precursor of these pollutants. We have now edited the text to clarify this [A]:**

[A]

**Before:**
*In the US city of Houston, Texas – the fifth most populous metropolitan region in the US (United States Census Bureau, 2022) – $NO_2$ pollution is a major concern (Mazzuca et al., 2016).*

**After:**
*In the US city of Houston, Texas – the fifth most populous metropolitan region in the US (United States Census Bureau, 2022) – $NO_2$ is a major concern (Mazzuca et al., 2016) due to its role as a precursor of the formation of $O_3$ and $PM_{2.5}$. While $NO_2$ itself nor $PM_{2.5}$ exceed their respective US EPA National Ambient Air Quality Standards (NAAQS), Houston is in moderate nonattainment of the 8-hour Ozone (2015) NAAQS.*

[3] Line 84 – What do you mean by "remote-sensing columns"? Satellite?

**We apologize for the vagueness, we meant both satellite and aircraft observations of $NO_2$ columns and have clarified the text in response to this comment [A]:**

[A]

**Before:**
*CAMx has been used extensively to investigate Texas air quality by leveraging model input data created by the Texas Commission on Environmental Quality (TCEQ) for air quality planning (Ge et al., 2021; Goldberg et al., 2022) with strong performance compared to remote-sensing column concentrations in Texas (Goldberg et al., 2022; W. Li et al., 2023; Soleimanian et al., 2023).*

**After:**
*CAMx has been used extensively to investigate Texas air quality by leveraging model input data created by the Texas Commission on Environmental Quality (TCEQ) for air quality planning (Ge et al., 2021; Goldberg et al., 2022) with strong performance compared to satellite and aircraft remote-sensing column concentrations in Texas (Goldberg et al., 2022; W. Li et al., 2023; Soleimanian et al., 2023).*

[4] Line 159 – This is confusing. Why would you compare Pandora to re-gridded GCAS and not the native resolution GCAS?

**Sorry about the confusion, this was a mistake on our part in the original manuscript, we do compare the Pandora data to the native resolution GCAS. The text has been updated to reflect this [A]:**

[A]

**Before:**
*Spatially, we identify the CAMx grid cell in which each Pandora instrument is located and only consider GCAS measurements that were regridded to these grid cells*

**After:**
*Spatially, we restrict our comparison to only the GCAS pixels that contain Pandora instruments.*

[5] Line 160 – 30 minutes seems like a long time given that direct sun Pandora data are more frequent and could be matched more closely. What is the variability in the data over that window and is it really reasonable to use such a long time period?

**We agree that this 30 minute period (15 minutes on either end of the overpass) is somewhat long; however, we chose 30 minutes as an upper bound cut-off, in reality, most of the Pandora data we present are within a few minutes of the GCAS / TROPOMI overpasses. We could choose a tighter window (e.g., 20 minutes or 15 minutes) but this would have a negligible impact on the comparison as most of the Pandora observations are taken very close to the overpass time. To quantify the impact of this long window we calculated the percentage of Pandora matches to both GCAS and TROPOMI that were within a 20 and 15 minute window instead and have added new text to summarize this and demonstrate that the length of the window has little impact on the overall conclusions [A]:**

[A]

**New Text:**

*While we choose this 30-minute window as an upper-bound cut-off, 96% and 90% of all Pandora closest matches occur within a 20- and 15-minute window of GCAS overpasses, respectively, indicating that this choice of window will have a minimal impact on our results.*

**New Text:**

*While we choose this 30-minute window as an upper-bound cut-off, 100% and 97% of all Pandora closest matches occur within 20 and 15 minutes of TROPOMI overpasses, respectively, indicating that this choice of window will have little impact on our results.*

[6] Line 190 – For comparison to TROPOMI, you need to regrid the model to the coarser TROPOMI resolution of 3.5x5.5 km2, otherwise the comparison will certainly look poor.

**In section 3.4 we have done this already and found generally comparable performance albeit improved correlation when comparing results at a coarser resolution.**

[7] Line 242 – Can you explain what you mean by "Third, we reprocess link-based on-road mobile emissions for the higher resolution domains."? What is the native resolution of the emissions? Is this some sort of down-scaling procedure?

**We apologize for the confusion. These emissions are derived from vector links that can be scaled to different resolutions. These link-based emissions correspond to lines that can be overlayed over grid cell polygons and thus emissions can be derived for different grid resolutions. In response to this comment we have added new text to the manuscript [A]. We have explicitly included emission resolutions in Table 3 – that is attached to the end of this document for convenience – and provide further detail in response to a minor comment [9] from RC1.**

[A]

**New Text:**

*Specifically, all point sources are geo-located to the grid cell containing the source. On-road mobile source emissions and shipping emissions were provided for individual links which we allocated to 444 m grid cells, and are based on known roadway networks, ship tracks, and traffic patterns. Airport and railyard emissions were allocated to 444 m grid cells within the property boundary. Other sources retained the 4 km grid resolution provided by the TCEQ.*

[8] Line 249 – High degree of uncertainty does not seem like a good reason to remove fire or lightning emissions. Does the model suggest that they are unimportant in Houston and thus you remove them to save computational expense?

**To be clear, fire and lightning $NO_x$ emissions were included in the 36, 12, and 4 km outer domains, but not the 1.33 km and 444 m domains. The fine grids (1.33 km and 444 m) cover the urbanized Houston area. Wildfires within the Houston area are rare, and satellite-derived wildfire emission inventories can inadvertently mis-classify the heat**

**signature of industrial flares as wildfires, so we excluded wildfire emissions in these fine grids. Lightning emissions are associated with convective clouds that would obscure detections by TROPOMI / Pandora / GCAS. In response to your comment, we have changed the text to better reflect this [A]:**

[A]

**Before:**
*Considering the limited extent of the high- resolution CAMx domains and large degree of uncertainty with both the fire and LNOx emissions, we excluded these two emission sources from the finer resolution domains.*

**After:**
*Considering* that wildfires in the Houston area are rare and that $LNO_x$ emissions are associated with convective clouds that obscure remote sensing column observations, we excluded these two emission sources from the finer resolution domains (the 1.333 and 0.444 km domains) but included them in the larger domains. These two sources represent a small fraction of emissions in the local Houston area that is the primary focus of the finer resolution simulations.*

[9] Line 256 – Why are you including GEOS-CF in this comparison? You haven't given us a motivation to do this.

**We agree that we were missing motivation for including GEOS-CF in our comparison. We include GEOS-CF to characterize differences between a global simulation (GEOS-CF) and our regional WRF-CAMx modeling and due to its relevance to TEMPO as TEMPO NO$_2$ will be using GEOS-CF [A]:**

[A]

**New Text:**
*We include $NO_2$ data from GEOS-CF – that will be used for processing $NO_2$ remote-sensing observations from the NASA Tropospheric Emissions: Monitoring of Pollution (TEMPO) mission – to characterize differences between a global simulation and our regional WRF-CAMx modeling.*

[10] Line 332 – Do you think the model successfully captures horizontal advection and vertical mixing in Houston? How could errors in transport relate to the model underestimate in NO2 columns? Does the model perform significantly differently on days with slower or faster wind speeds?

**We thank you for these questions. Based on our evaluation of meteorology we do not see any systematic biases in model performance associated with wind speed / direction. Explicitly, biases on non-cloudy days in wind direction were on average -5° and ranged from -25° to +11° and wind speed biases were on average -0.02 m/s and ranged from -0.93 to +0.68 m/s. There are periods when WRF imperfectly captures wind speed and direction; however, on the daily time-scale model performance appears to be relative stable. Given**

this, we believe the model performs well in terms of horizontal wind speed. In response to your question on how errors in transport could relate to model underestimate, we suggest that non-systematic differences would degrade correlation between simulated and observed $NO_2$ columns and add this to the text [A].

Investigations into vertical mixing would be interesting and we believe this could warrant further study, but this is beyond the scope of this project. Liu et al. 2023 has already done this for the same timeframe and Houston domain using various WRF configurations, albeit with a different chemical transport model; we now cite this paper. Given that we have a low bias both at the surface (NMB=-59%) and in the column (NMB=-25.5%) we believe that vertical mixing is not the primary culprit for the bias in this study.

In response to your question regarding how errors in transport could relate to model underestimates, if vertical mixing was biased high, this could explain the worse performance at the surface as vertical wind would remove $NO_2$ faster from the surface in the WRF simulation than in reality. If vertical mixing was biased low, that would mean in the current simulation, $NO_2$ concentrations would be accumulating at the surface-level and simulated surface-level $NO_2$ would be higher than expected. Subsequently, the already low biased surface $NO_2$ should be even lower if this is the case which seems unlikely. To address this comment, we have added some new text in the discussion suggesting that this is an area of further investigation [B].

In response to your last question, excluding cloudy days, the two days with the highest winds (9/10 and 9/11) have comparable correlations and biases in both wind speed ($R^2$=0.5 and 0.32; MAE=0.98 and 1.35 m/s) and wind direction ($R^2$=0.38 and 0.68; MAE=17.0° and 14.0°) to the other simulation days for both wind speeds ($R^2$ ranges from 0.1 to 0.41; MAE ranges from 0.96 to 1.25) and wind directions ($R^2$ ranges from 0.38 to 0.78; MAE ranges from 19° to 32°). We have added a sentence addressing this in the text [C].

[A]

**New Text:**
*We generally find good agreement in the WRF simulated meteorology (Table S5-8 and Fig. S2-11) however, the non-systematic differences on the order of 20° would likely degrade correlation between observed and simulated $NO_2$ columns.*

[B]

**New Text:**
*Although we evaluate the performance of WRF at the surface, we do not consider vertical advection and the vertical mixing scheme in WRF merits further investigation. A previous study by Liu et al., (2023) has investigated this for the TRACER-AQ campaign in Houston, albeit with a different chemical transport model, and found generally good agreement in potential temperature but an underestimate of ozone in the free troposphere. Given the worse performance of WRF-CAMx at the surface (NMB=-59%) than for the columns (-26%), if the vertical mixing*

*scheme has poor performance, we suspect it to be due to overmixing leading to the rapid removal of surface-level NO2.*

[C]

**New Text:**
*We do not find notable differences in correlation or error on windier days than on calmer days (Table S5-S6).*

[11] Line 348 – Again, any windspeed or direction dependence? What about hour of day dependence?

**We thank you for this comment, we have endeavoured to address concerns involving windspeed and direction in our response to the previous comment [10]. To address your comment regarding hour of day dependence, we have generated figures for the supplement to investigate performance at different hours of the day (Fig. S2-S13). We focused on the early morning and afternoon periods – although we examined performance across all hours – and did not find any notable dependence. The one exception is on cloudy days we have poorer performance; however, since there are limited observations on these days to begin with, they have little influence on our overall results. Beyond these figures, we add new text to mention the poor performance during cloudy periods [A].**

[A]

**New Text:**
*Generally, meteorological conditions simulated by WRF agree with ground-level observations especially on the more data rich non-cloudy days that are the most important for our intercomparison.*

**Community Commenter #1:**

**Minor Comments**

[1] Line 20. Define which correlation statistic is used here. Is it Pearson-R?

Line 21. NMB abbreviation is used in the abstract but is not defined. Does it mean 'normalized mean bias' or 'normalized median bias'

**We thank the community commenter for their useful responses. In response to these and other comments throughout, we have added a section to the supplemental that defines all statistics discussed in this manuscript which we include below for convenience [A]. Additionally, we have added text to refer to this new section in the main manuscript [B].**

[A]

*Section S2. Equations for statistics referenced in the main text*

*$R^2$ (Pearson-R squared):* $R^2 = 1 - \frac{\Sigma(y_{obs} - y_{prd})^2}{\Sigma(y_{obs} - \overline{y_{obs}})^2}$

*NMB (Normalized Mean Bias):* $NMB = \frac{\Sigma(y_{prd} - y_{obs})}{\Sigma y_{obs}}$

*MBE (Mean Bias Error):* $MBE = \frac{1}{n}\Sigma(y_{prd} - y_{obs})$

*MAE (Mean Absolute Error):* $MAE = \frac{1}{n}\Sigma|y_{prd} - y_{obs}|$

*$\sigma$ (Standard deviation of the residuals):* $\sigma = \sqrt{\frac{\Sigma(r - \bar{r})^2}{n}}$

*$y_{obs}$: Data from the dataset suspected to be closest to truth (Pandora > GCAS > TROPOMI > CAMx)*
*$y_{prd}$: Data from the dataset suspected to be furthest from truth*
*$\overline{y_{obs}}$: Mean of $y_{obs}$*
*$r$: $y_{prd} - y_{obs}$*
*$\bar{r}$: Mean of $r$*
*$n$: Number of observations*

**New Text:**
*All statistical measures (e.g., $R^2$) are defined in the supplement.*

[2] Line 114-115. "Pan #59 at Aldine, #61 at La Porte, and #25 and University of Houston were chosen..."
59 is not in Table 1. 61 is not at La Porte but at Aldine, according to Table 1.

**Thank you for alerting us of this – the text was incorrectly numbered but this has now been fixed to align with the Table.**

[3] Table 1. I would include here also the altitude, as it is a reason for not using P188.

**We thank you for this suggestion, but we have decided to not include this in the table as Pan 188 was the only site with dramatically different altitude.**

[4] Line 172. "We download the publicly available data (https://data-portal.s5p-pal.com/products/no2.html) "

In fact, only data processed with processor version 2.3.1, with data up to November 2021, can be found here. The latest reprocessed version 2.4.0 (and 2.5.0 for measurements since March 2023) are currently available from the Copernicus Data Space Ecosystem (https://dataspace.copernicus.eu/)

**Thank you for sharing this updated link with us, we have now included it in the manuscript.**

[5] Line 186. "A comparison between TROPOMI version 2.4.0 and a MAX-DOAS network found that in moderately polluted locations, TROPOMI had a median bias of -35% (Lambert et al., 2023). "

Not only MAX-DOAS, but also Pandora measurements from PGN are included in this quarterly updated validation report. Given that Pandora measurements are used in the current paper, it is logical to mention these results as well.

**Thank you for alerting us that PGN measurements are included in the quarterly validation reports. Given that these are relevant to our comparison, we have now included this information in the main text:**

**New Text:**

*A comparison between TROPOMI version 2.4.0 and a MAX-DOAS network found that in moderately polluted locations, TROPOMI had a median bias of -35%; a comparison between TROPOMI version 2.4.0 and PGN found a median bias of -18% over polluted stations (Lambert et al. 2023).*

[6] Line 189-190. What kind of regridding approach is applied?

**Thank you for this question, we follow the approach used in a previous study (Goldberg et al., 2021). Briefly, we "push" data to a finer grid. If a center point of the grid cell is within**

**the bounds of a TROPOMI pixel, the value of the grid cell is the value of the TROPOMI pixel. The full reference to this paper has been added in the references section at the end of this document.**

[7] Line 196-198.  More detail is needed here. Which TROPOMI version is used? How exactly are the total and tropospheric values calculated? By calculating new total and tropospheric AMF and in this way recalculating from the available tropospheric and total VCD available in the TROPOMI file?

Note also that in the operational product, there are two total column variables. Normally only the 'summed column' is recommended to be used.

Also, there is already a stratospheric NO2 available in the TROPOMI data files. Why was this not used to correct the GCAS and WRF-CAMx data?

**We thank you for these questions and agree that the text can be further clarified to address these concerns. We use TROPOMI version 2.4.0 to calculate the correction factor and have new text to mention this [A]. We follow the TROPOMI Users Guide, section 8.8 to calculate the total and tropospheric values** (*Sentinel-5P TROPOMI Level-2 Product User Manual - Nitrogen Dioxide*) **and have provided more detail on our method in the manuscript [B]. In response to your last question, we do in fact use the stratrospheric NO$_2$ from TROPOMI to estimate the total column NO$_2$ of GCAS and CAMx when comparing to Pandora and have now included new text that mentions this [B].**

[A]

**New Text:**
*Using WRF-CAMx vertical profile information we calculate both a total and tropospheric NO$_2$ column from TROPOMI v2.4.0 measurements using a new AMF derived from the WRF simulation and we difference the total and tropospheric values to calculate a stratospheric NO$_2$ column component from TROPOMI.*

[B]

**New Text:**
*Specifically, we use the AMF from the CAMx simulation to calculate a tropospheric NO$_2$ column from TROPOMI; following the TROPOMI users guide, we multiply the total averaging kernel by the ratio of the total air mass factor to the tropospheric air mass factor. We difference the total column NO$_2$ from TROPOMI with the tropospheric column to estimate a stratospheric NO$_2$ column correction factor that we apply to GCAS and CAMx when we compare them to Pandora; this corresponds to a value of $3.0 \times 10^{15}$. For GCAS, we apply an additional correction factor to account for the NO$_2$ column above the aircraft and below the tropopause. We calculate the column of the 3 highest levels of CAMx that correspond to 9400-18100 m above sea-level – that is roughly the same height that GCAS flies at of around 9100 m – and apply this to the GCAS results; this corresponds to a value of $0.57 \times 10^{15}$ molecules cm$^{-2}$. For all results that include comparison to Pandora we present total NO$_2$ columns; for results where we only intercompare*

*GCAS, TROPOMI, and CAMx we compare the tropospheric column. All statistical measures (e.g., $R^2$) are defined in the supplement.*

[8] Line 309. r^2 and NMB are used but not introduced. Is Pearson-R used? What is the exact mathematical definition of NMB?

I see later in the text 'normalized mean bias' appearing, but that does not yet explain how exactly it is normalized (using only reference data or e.g., the mean of reference and satellite data), and if the normalization happens before or after the averaging.
When using relative measures, I recommend using median rather than the mean, to prevent small numbers in the denominator to dominate.
I recommend also to add a statistic on the dispersion. E.g., the standard deviation of the differences.
Fig. 2. MBE, NMB are used in the figures but are not defined.

**We agree with you that further clarity on the statistical measures was needed in the original manuscript. In response to your minor comments [1] we have added a new section to the supplement in which statistical measures are defined. We thank the commenter for their suggestion on using median biases as opposed to mean biases; however, we have decided to keep using mean biases. Based on this comment we have now added a measure of dispersion – the standard deviation – that is included on the figures in the updated manuscript.**

[9] Also there is a typo in the unit: this should be molecules cm-2 10^16, not molecules cm-2 10^(-16). I recommend to add a dispersion measure like the standard deviation of the differences.

**Thank you for catching this typo, we have noticed this typo in a few locations, and we have now updated this throughout the manuscript.**

[10] Line 312. "but there was a negative bias (NMB=-22.8%) in v2.4.0. "

Here, it is relevant to compare with the TROPOMI vs Pandora results on the validation server (https://mpc-vdaf-server.tropomi.eu).
Results using Pandora #25 (university of Houston; PGN location name HoustonTX) are at

https://mpc-vdaf-server.tropomi.eu/no2/no2-offl-pandora/houston-tx
Results using Pandora #58 are at https://mpc-vdaf-server.tropomi.eu/no2/no2-offl-pandora/la-porte-tx-gsfc058
While it is not fully clear to me what is the meaning of MBE and NMB, the numbers seem broadly consistent with the bias statistics obtained from the validation server, despite the fact that the sampling is not the same (as on the validation server no subsampling to the airborne measurements is done). This indicates that the sample taken in the current paper is a representative set.

Note that on the server, the filter qa_value>75% is not applied yet (only qa_value>50%), so you'll first have to download the data and apply this more strict filter yourself to obtain exactly

the numbers below. This setting was taken to have more pixels also for the sites where the stratospheric contribution is dominant.

La Porte TX P58:

mean difference=-2.5 10^15 molec/cm2

median difference =-2.2 10^15 molec/cm2

mean relative difference (i.e., mean of (SAT-REF)/REF)=-21%

median relative difference (i.e., median of (SAT-REF)/REF)=-24%
Houston TX P25:

mean difference=-2.3 10^15 molec/cm2

median difference =-1.9 10^15 molec/cm2

mean relative difference =-21%

median relative difference=-23%
Finally, also the comparison with La Porte TX P63 (mentioned in Table 1 but not used in the paper) is available on the validation server. However, in this case the bias is quite different, due to Pandora values being lower (about 2 Pmolec cm-2 lower when calculating on an overlapping time range 2021-10-07 to 2022-01-22).

**We thank you for comparing our results to the results on the validation server and we are excited to see that there is generally comparable performance. While we believe that a comparison between our results and those presented on the validation server is beyond the scope of this manuscript, we are interested investigating this at a later date and encourage the commenter to contact us. We also note that this time range does not overlap with the TRACER AQ campaign which ran in September 2021.**

[11] In general, it would be interesting if the authors added a consistency check of the Pandora instruments that are at the same location, perhaps in section 2.1.

**We agree with you that this would be interesting. Generally we found comparable performance across collocated Pandora instruments – with the exception of #188 due to the large altitude difference. We have chosen to limit the Pandora results presented in this paper only to the three chosen instruments to avoid any confusion.**

Line 607. 'both outperform version 2.4.0'. The bias of NASA MINDS and v2.3.1 is indeed slightly better, but the correlation of NASA MINDS is lower than that of 2.4.1. Also, it would be interesting to look at a dispersion measure (e.g., standard deviation of the differences).

**We agree with you that the overall performance was not necessarily better, just the bias. This has now been clarified in updated text [A]:**

[A]

**New Text:**
*When comparing different versions of TROPOMI we find differences between the v2.3.1, v2.4.0, and NASA MINDS product and find that the MINDS ($r^2$=0.69 and NMB=-18.2%) and version 2.3.1 ($r^2$=0.72 and NMB=-18.3%) products – with the CAMx AMF – performs comparably but both outperform version 2.4.0 considering bias albeit with slightly worse correlation*

Line 643. "It is common to consider TROPOMI measurements as accurate representation of NO 2 column concentrations; however, if we had done so in this study, we would have failed to identify the substantial negative bias"

Here, it is useful to remind that a negative bias has been mentioned in earlier literature (e.g., Verhoelst et al, 2021) and in the quarterly issued operational validation reports (available at https://mpc-vdaf.tropomi.eu/). This information is therefore already available to the user.

**Thank you for this comment, we agree with this point. Negative biases in TROPOMI in urban areas have previously been documented and this information is publicly available. Given this, we have updated the text in the manuscript:**

**Before:**
*This analysis benefitted from three independent measurement datasets (i.e., Pandora, TROPOMI, and GCAS) that were critical to isolate the negative biases in TROPOMI and CAMx.*

**After:**
*This analysis benefitted from three independent measurement datasets (i.e., Pandora, TROPOMI, and GCAS) that were critical to isolate the negative biases in TROPOMI and CAMx although we note that negative biases in TROPOMI have been mentioned in earlier literature (e.g., Verhoelst et al, 2021) and in the quarterly issued operational validation reports (available at https://mpc-vdaf.tropomi.eu/).*

Line 648. In the acknowledgments, there should as a minimum appear: 'contains modified Sentinel data'.

An example sentence could be 'This work contains modified Copernicus Sentinel-5 Precursor

data processed by KNMI and post-processed by ... (ending with the name of your research institute).

**Thank you telling us about this, we have now updated the acknowledgements.**

**Figures and tables referenced in responses:**

[Figure]

**Figure 1: Comparison of Pandora total column NO₂ to GCAS using CAMx-based AMFs (A), TROPOMI v2.4.0 using CAMx-based AMFs (B), and CAMx (C) and GCAS (D) and TROMPOMI v2.4.0 (E) with their operational AMFs. Tropospheric columns from GCAS and CAMx are bias corrected with a TROPOMI-derived stratospheric column factor as discussed in the methodology. Data from all possible overpasses coincident within 15 minutes of a Pandora observation are considered. GCAS flight times generally ranged**

from 8:00 AM-4:00 PM CDT. TROPOMI overpasses occurred around 1:30 PM local time. Color coding indicates which of the Pandora instruments NO₂ column concentrations are being compared against indicated in the legend in subplot A, but statistics are presented across all locations. Map of Pandora instrument sites in urban Houston (F). Bias between the three datasets and Pandora across GCAS flight days (G) with the overall average daily bias indicated above the points for all three datasets. The data are color coded based on the observed or simulated source that is being compared against Pandora measurements. © OpenStreetMap contributors 2023. Distributed under the Open Data Commons Open Database License (ODbL) v1.0.

[Figure]

**Figure 3: Comparison between Pandora measurements and TROPOMI observations using the CAMx AMF for version 2.3.1 (A), 2.4.0 (B), and NASA MINDS (C) and the same respective versions using the operational AMF (D-F). Data from all possible overpasses coincident within 15 minutes of a Pandora observation are considered with one exception: data from September 11th, 2021, was missing from the NASA MINDS product and so values in plots C and F exclude this day.**

[Figure]

**Figure 7: Comparison of GCAS (A), TROPOMI (B), and CAMx (C) to Pandora on weekdays and of GCAS (D), TROPOMI (E), and CAMx (F) to Pandora on weekends. Data from all possible overpasses coincident within 15 minutes of a Pandora observation are considered. GCAS flight times generally ranged from 8:00 AM-4:00 PM CDT. TROPOMI overpasses occurred around 1:30 PM local time.**

[Figure]

**Figure 9: Diurnal patterns in Total NO₂ columns (top) averaged across the three Pandora sites and 10 flight days from CAMx (green), GCAS (blue), Pandora (red), and GEOS-CF (black). Diurnal patterns in surface-level NO₂ concentrations (bottom) in downtown Houston for CAMx and GEOS-CF averaged across the 10 flight days and across all monitors in Harris County for AQS surface-level monitors (red).**

**Table 3: CAMx 444 × 444 m² domain-wide summary of average September weekday emissions by sector in units of tons per day (tpd).**

| Emission Sector | Spatial Resolution | NOₓ (tpd) | VOC (tpd) |
|---|---|---|---|

| | | | |
|---|---|---|---|
| EGUs | Point locations | 25.5 | 0.2 |
| On-road mobile | Line source | 70.9 | 34.7 |
| Railyards | 444 m gridded | 4.2 | 0.3 |
| Shipping | Line source | 63.9 | 4.3 |
| KIAH airport | 444 m gridded | 6.4 | 0.8 |
| KHOU airport | 444 m gridded | 1.8 | 0.4 |
| Other | | | |
| Off-road mobile | 4 km gridded | 33.1 | 31.4 |
| Non-EGU Point Sources | Point locations | 47.9 | 27.8 |
| Oil and Gas | 4 km gridded | 0.2 | 0.0 |
| Area | 4 km gridded | 92.8 | 623.2 |
| MEGAN biogenic | 444 m gridded | 25.9 | 319.7 |

**Table S1: WRF physics options and data sources**

| WRF Option | Option Selected |
|---|---|
| Analysis Data | 0.25° GDAS (IC/BCs and analysis nudging on the 36 and 12 km domains) |
| Microphysics | Thompson |
| Longwave Radiation | Rapid Radiative Transfer Model (RRTMG) |
| Shortwave Radiation | RRTMG |
| Surface Layer Physics | Revised MM5 surface layer scheme |
| LSM | Noah |
| PBL scheme | Yonsei University (YSU) |
| Cumulus scheme | Multi-Scale Kain-Fritsch (MSKF) on 36/12 km; none for 4/1.333/0.444 km |

**Table S5: Comparison of observed (OBS) and simulated (WRF) wind direction and associated statistics. Red shading indicates days with limited GCAS observations due to cloud coverage. The statistical measures of mean bias error (MBE), mean absolute error (MAE), and Pearson-R squared ($R^2$) are computed using the astropy circular statistics python module (https://docs.astropy.org/en/stable/stats/circ.html).**

| Date | WRF Dir (°) | OBS Dir (°) | $R^2$ | MBE | MAE | N |
|---|---|---|---|---|---|---|
| 09/01 | 184.0 | 180.0 | 0.04 | -1.0 | 36.0 | 130 |
| 09/03 | 136.0 | 185.0 | -0.04 | -49.0 | 60.0 | 137 |
| 09/08 | 13.0 | 17.0 | 0.57 | -10.0 | 29.0 | 137 |
| 09/09 | 20.0 | 37.0 | 0.78 | -11.0 | 24.0 | 149 |
| 09/10 | 81.0 | 75.0 | 0.38 | 6.0 | 17.0 | 149 |
| 09/11 | 89.0 | 92.0 | 0.68 | -3.0 | 14.0 | 151 |
| 09/23 | 63.0 | 88.0 | 0.3 | -24.0 | 28.0 | 118 |
| 09/24 | 87.0 | 97.0 | 0.47 | -8.0 | 20.0 | 137 |
| 09/25 | 70.0 | 59.0 | 0.57 | 9.0 | 24.0 | 114 |
| 09/26 | 97.0 | 96.0 | 0.8 | 0.0 | 19.0 | 117 |
| All Days | 82.0 | 87.0 | 0.76 | -8.0 | 26.0 | 1339 |
| Not Cloudy Days | 67.0 | 73.0 | 0.73 | -5.0 | 21.0 | 1072 |

**Table S6: Comparison of observed (OBS) and simulated (WRF) wind speed and associated statistics. Red shading indicates days with limited GCAS observations due to cloud coverage. The statistical measures of mean bias error (MBE), mean absolute error (MAE), and Pearson-R squared ($R^2$) are defined at the end of this supplement in section S2.**

| Date | WRF Spd (m/s) | OBS Spd (m/s) | $R^2$ | MBE | MAE | N |
|---|---|---|---|---|---|---|
| 09/01 | 3.47 | 2.93 | 0.07 | 0.55 | 1.51 | 145 |
| 09/03 | 3.46 | 3.17 | 0.0 | 0.29 | 1.8 | 151 |
| 09/08 | 3.46 | 2.79 | 0.1 | 0.68 | 1.25 | 150 |
| 09/09 | 3.7 | 3.76 | 0.23 | -0.06 | 0.94 | 152 |
| 09/10 | 4.6 | 4.69 | 0.5 | -0.09 | 0.98 | 149 |
| 09/11 | 4.11 | 5.04 | 0.32 | -0.93 | 1.35 | 151 |
| 09/23 | 3.14 | 3.2 | 0.2 | -0.06 | 1.17 | 130 |
| 09/24 | 3.32 | 3.77 | 0.41 | -0.45 | 0.96 | 142 |
| 09/25 | 2.73 | 2.66 | 0.25 | 0.07 | 0.94 | 135 |
| 09/26 | 3.1 | 3.29 | 0.36 | -0.2 | 0.99 | 131 |
| All Days | 3.53 | 3.55 | 0.26 | -0.02 | 1.2 | 1436 |
| Not Cloudy Days | 3.55 | 3.68 | 0.37 | -0.13 | 1.08 | 1140 |

**Table S7: Comparison of observed (OBS) and simulated (WRF) temperature and associated statistics. Red shading indicates days with limited GCAS observations due to cloud coverage. The statistical measures of mean bias error (MBE), mean absolute error (MAE), and Pearson-R squared ($R^2$) are defined at the end of this supplement in section S2.**

| Date | WRF Temp (K) | OBS Temp (K) | $R^2$ | MBE | MAE | N |
|---|---|---|---|---|---|---|
| 09/01 | 305.5 | 304.2 | 0.0 | 1.3 | 2.28 | 159 |
| 09/03 | 305.0 | 303.0 | 0.0 | 1.99 | 2.61 | 158 |
| 09/08 | 304.5 | 304.1 | 0.75 | 0.42 | 1.16 | 160 |
| 09/09 | 305.1 | 304.4 | 0.78 | 0.66 | 1.15 | 160 |
| 09/10 | 303.1 | 303.2 | 0.76 | -0.1 | 0.93 | 158 |
| 09/11 | 302.5 | 302.3 | 0.74 | 0.18 | 0.99 | 157 |
| 09/23 | 298.4 | 298.6 | 0.69 | -0.26 | 1.14 | 147 |
| 09/24 | 299.2 | 299.3 | 0.72 | -0.11 | 1.02 | 150 |
| 09/25 | 299.5 | 299.9 | 0.81 | -0.33 | 1.02 | 148 |
| 09/26 | 300.0 | 300.1 | 0.69 | -0.02 | 1.18 | 150 |
| All Days | 302.4 | 302.0 | 0.71 | 0.39 | 1.35 | 1547 |
| Not Cloudy Days | 301.6 | 301.5 | 0.86 | 0.06 | 1.07 | 1230 |

**Table S8: Comparison of observed (OBS) and simulated (WRF) water vapor mixing ratio (WVMR) and associated statistics. Red shading indicates days with limited GCAS observations due to cloud coverage. The statistical measures of mean bias error (MBE), mean absolute error (MAE), and Pearson-R squared ($R^2$) are defined at the end of this supplement in section S2.**

| Date | WRF WVMR (g/kg) | OBS WVMR (g/kg) | $R^2$ | MBE | MAE | N |
|---|---|---|---|---|---|---|
| 09/01 | 17.5 | 19.4 | 0.26 | -1.92 | 1.97 | 99 |
| 09/03 | 17.7 | 18.6 | 0.21 | -0.88 | 1.22 | 98 |
| 09/08 | 12.3 | 12.2 | 0.78 | 0.09 | 1.31 | 100 |
| 09/09 | 10.7 | 13.8 | 0.28 | -3.07 | 3.11 | 100 |
| 09/10 | 9.1 | 10.7 | 0.67 | -1.51 | 1.6 | 98 |
| 09/11 | 10.6 | 12.6 | 0.21 | -2.03 | 2.28 | 97 |
| 09/23 | 6.0 | 7.6 | 0.54 | -1.55 | 1.58 | 97 |
| 09/24 | 7.4 | 8.8 | 0.32 | -1.34 | 1.46 | 100 |
| 09/25 | 8.2 | 9.7 | 0.54 | -1.51 | 1.53 | 98 |
| 09/26 | 10.8 | 11.6 | 0.48 | -0.76 | 0.87 | 100 |
| All Days | 11.1 | 12.5 | 0.85 | -1.45 | 1.69 | 987 |
| Not Cloudy Days | 9.4 | 10.9 | 0.64 | -1.46 | 1.72 | 790 |

**Figure S2: Comparison of observed (OBS) and simulated (WRF) wind at 9am and 1pm CST across sixteen ground-level monitors on September 1, 2021. © OpenStreetMap contributors *2023*. Distributed under the Open Data Commons Open Database License (ODbL) v1.0.**

[Figure]

**Figure S3: Comparison of observed (OBS) and simulated (WRF) wind at 9am and 1pm CST across sixteen ground-level monitors on September 3, 2021. © OpenStreetMap contributors *2023*. Distributed under the Open Data Commons Open Database License (ODbL) v1.0.**

[Figure]

[Figure]

**Figure S4: Comparison of observed (OBS) and simulated (WRF) wind at 9am and 1pm CST across sixteen ground-level monitors on September 8, 2021. © OpenStreetMap contributors *2023*. Distributed under the Open Data Commons Open Database License (ODbL) v1.0.**

[Figure]

**Figure S5: Comparison of observed (OBS) and simulated (WRF) wind at 9am and 1pm CST across sixteen ground-level monitors on September 9, 2021. © OpenStreetMap contributors *2023*. Distributed under the Open Data Commons Open Database License (ODbL) v1.0.**

[Figure]

**Figure S6: Comparison of observed (OBS) and simulated (WRF) wind at 9am and 1pm CST across sixteen ground-level monitors on September 10, 2021. © OpenStreetMap contributors *2023*. Distributed under the Open Data Commons Open Database License (ODbL) v1.0.**

[Figure]

**Figure S7: Comparison of observed (OBS) and simulated (WRF) wind at 9am and 1pm CST across sixteen ground-level monitors on September 11, 2021. © OpenStreetMap contributors *2023*. Distributed under the Open Data Commons Open Database License (ODbL) v1.0.**

[Figure]

**Figure S8: Comparison of observed (OBS) and simulated (WRF) wind at 9am and 1pm CST across sixteen ground-level monitors on September 23, 2021. © OpenStreetMap contributors *2023*. Distributed under the Open Data Commons Open Database License (ODbL) v1.0.**

[Figure]

**Figure S9: Comparison of observed (OBS) and simulated (WRF) wind at 9am and 1pm CST across sixteen ground-level monitors on September 24, 2021. © OpenStreetMap contributors *2023*. Distributed under the Open Data Commons Open Database License (ODbL) v1.0.**

[Figure]

**Figure S10: Comparison of observed (OBS) and simulated (WRF) wind at 9am and 1pm CST across sixteen ground-level monitors on September 25, 2021. © OpenStreetMap contributors *2023*. Distributed under the Open Data Commons Open Database License (ODbL) v1.0.**

[Figure]

**Figure S11: Comparison of observed (OBS) and simulated (WRF) wind at 9am and 1pm CST across sixteen ground-level monitors on September 26, 2021. © OpenStreetMap contributors *2023*. Distributed under the Open Data Commons Open Database License (ODbL) v1.0.**

[Figure]

**Figure S12: Hourly CAMx (7 AM-5 PM CST) surface NO₂ plotted against observed surface NO₂ across all TCEQ CAMS sites within Houston for all days with GCAS flight measurements during the August 30-September 27, 2021 modeling period.**

[Figure]

**Figure S13. MDA8 CAMx simulated and observed surface-level ozone across all TCEQ CAMS sites within Houston for all days with GCAS flight measurements during the August 30-September 27, 2021 modeling period.**

[Figure]

**References**

[revised manuscript text omitted]

---

## Referee Report (RR1)

**General Comments**

The authors have improved the manuscript and addressed many of the comments. The additional comparisons of temperature, windspeed, and wind direction against 16 ground monitors are useful. I still have two major comments:

1. The authors say there is good agreement also against the surface observations, but how do I know that their agreement is good? Table S6 gives an $R^2$ for wind direction of 0.26 which is not very good. I would have preferred to see scatterplots rather than (or in addition to) tables to determine whether the $R^2$ values were being driven by outliers or overall model performance. The authors should better discuss the meaning of their meteorological comparison statistics and how any biases could impact their results. This could be a very positive discussion that could inform future studies about what model biases are impacting emissions assessments. Currently the manuscript is a little confusing about what is 'good agreement' and what is a meaningful bias that needs to be discussed.

2. I am still concerned by the lack of evaluation of model vertical structure. As the model has a strong negative bias in $NO_2$, it is possible this is due to errors in wind direction (as the authors suggest) but also to potential excessive vertical mixing. As vertical structure is essential to successful satellite interpretation, I do not agree that comparisons between vertical profiles of temperature, RH etc is out of the scope of this project. I do not see how this is different than the comparison done with the surface stations. I would again strongly suggest the authors add an evaluation against the data from the TRACER-AQ ozonesondes (https://www-air.larc.nasa.gov/cgi-bin/ArcView/traceraq.2021?SONDE=1) that include temperature, pressure, windspeed, wind direction, RH, and of course ozone. Possibly the model output was not saved for this which then certainly would be burdensome to produce? If so, then a statement about how this could be done in the future (or with other datasets like HSRL-2 or TolNet) would be helpful.

**Specific Comments**
*Line numbers refer to the version with track changes.*

Line 170 – Please add the swath size for GCAS.

Line 237 – Previously I commented "Line 190 – For comparison to TROPOMI, you need to regrid the model to the coarser TROPOMI resolution of 3.5x5.5 km2, otherwise the comparison will certainly look poor." The response is as follows:

Fig. 2 – Is the model output at 15-min or hourly resolution? I read both in the previous text. It would be useful to know how variable Pandora is across the hour if using model hourly to know how much that matters.

**In section 3.4 we have done this already and found generally comparable performance albeit improved correlation when comparing results at a coarser resolution.**

Please add a reference to Section 3.4 about this then on line 237, otherwise as a reader I would be thrown off.

Line 446 - The authors state that there is a non-systematic difference in wind direction on the order of 20°, but then state two sentences later that there is no apparent systematic bias in meteorology. This is contradictory. Also, Tables S5-S6 give statistics for all days and non-cloudy days, but not windy vs. calm.

Line 550 – Can you give us some more information about the performance of the scheme used here (YSU?) compared to others?

Line 684 – Over how much of a larger extent?

Line 990 – The authors have discussed errors in meteorology throughout the paper including a bias of 20° in wind direction. I am not convinced errors are "minimal". It would be better to say that: with caveats that there are some

errors in meteorology, they are unlikely to fully explain the low $NO_2$ bias in the CAMx column and some of this bias may be attributable to underestimated emissions.

Data availability – the data availability statement does not include the TCEQ emissions inventory.

---

## Author Response (AR2)

**Response to the reviewer of:**

*An intercomparison of satellite, airborne, and ground-level observations with WRF-CAMx simulations of NO$_2$ columns over Houston, TX during the September 2021 TRACER-AQ campaign*

We thank the reviewer for their additional round of comments on the manuscript titled above. In this document we respond to their comments using the following notation: plain text indicates the reviewers' comments, **bolded text** indicates our responses, and *italicized text* indicates quotations from the original and updated manuscripts. For quotations, we will indicate whether they come from the original manuscript or the updated manuscript using "before" and "after", respectively; new text is indicated with red coloring. We have endeavoured to address the concerns of the reviewers and believe that these changes have improved the quality of the work.

**Anonymous Referee #2:**

**Major Comments:**

The authors say there is good agreement also against the surface observations, but how do I know that their agreement is good? Table S6 gives an R2 for wind direction of 0.26 which is not very good.

The authors should better discuss the meaning of their meteorological comparison statistics and how any biases could impact their results. This could be a very positive discussion that could inform future studies about what model biases are impacting emissions assessments. Currently the manuscript is a little confusing about what is 'good agreement' and what is a meaningful bias that needs to be discussed.

**Thank you for this comment. You are right that "good" agreement is not sufficiently objective. We have rephrased the description to highlight the performance metrics. The model is being used to represent the dispersion of pollution. We note that the GCAS retrievals show that the plumes are indeed being transported in the same direction in the model as in reality. Because the wind speed bias is so minimal, the speed of advection and associated dispersion should be accurate despite the low $R^2$. We revised the presentation of Table S6 to clarify that it refers to *wind speed* not *wind direction*. We find a correlation in simulated wind direction (Table S6) compared to observations of $R^2=0.76$ and a minimal bias of MBE=-8.0°. There is a mean absolute error of MAE=26° across all the ground-monitors, hours, and days. We agree that wind speed performance (Table S5) is indeed poorer ($R^2=0.26$) and that there is some unsystematic error (MAE=1.2 m/s) but almost no systematic error across all days (MBE=-0.02 m/s). We agree with your larger point – that we could more clearly discuss what we mean by *good performance* and refer to specific meteorological conditions – and have modified the text to address this:**

**Before:**

[revised manuscript text omitted]

I would have preferred to see scatterplots rather than (or in addition to) tables to determine whether the R2 values were being driven by outliers or overall model performance.

**We agree with you that whether $R^2$ values are being driven by outliers is an important consideration and a feature that should be discussed in the main text. While we do not include scatterplots to address this, the influence of outliers can be inferred from Figures S2-11. These figures reveal that poor performance in wind speed is possibly driven by values in the afternoon and the emergence of the Bay/Gulf breeze as there is generally improved agreement in wind-speed in the morning. Additionally, these figures reveal that simulated wind speeds at the downtown sites – that are most relevant to our conclusions regarding bias in the emissions inventory – generally perform better than the south-east sites near the Galveston Bay. We have now modified the text to address this:**

**New Text:**

*Considering wind speeds at 9am and 1pm (Fig. S2-11), it appears that observations in the afternoon degrade correlation compared to the morning and that, generally, simulated wind speeds are better correlated with observations in downtown Houston than in the south-eastern part of the domain near the Galveston Bay.*

I am still concerned by the lack of evaluation of model vertical structure. As the model has a strong negative bias in NO2, it is possible this is due to errors in wind direction (as the authors suggest) but also to potential excessive vertical mixing. As vertical structure is essential to successful satellite interpretation, I do not agree that comparisons between vertical profiles of temperature, RH etc is out of the scope of this project. I do not see how this is different than the comparison done with the surface stations. I would again strongly suggest the authors add an evaluation against the data from the TRACER-AQ ozonesondes (https://www-air.larc.nasa.gov/cgi-bin/ArcView/traceraq.2021?SONDE=1) that include temperature, pressure, windspeed, wind direction, RH, and of course ozone. Possibly the model output was not saved for this which then certainly would be burdensome to produce? If so, then a statement about how this could be done in the future (or with other datasets like HSRL-2 or TolNet) would be helpful.

**We thank you for this comment, and we agree that model vertical structure is an important consideration when evaluating model performance for column quantities, although we note that columns are likely less sensitive to vertical structure than ground-level measurements. To address this comment, we have briefly compared model vertical structure against the ozonesondes at five different sites representing different days and times that we have now added to the supplement as Figures S12-S16 (included in this document for convenience). We note that we do not include RH as this model output was not saved out except at the surface. We find that the model simulates temperature and pressure well across all five ozonesondes; however, vertical profiles in O₃ mixing ratio, wind speed, and wind direction are more mixed. For some sondes (Figure S16-S17) we find great agreement, especially below 1km; however, for others there does appear to be poorer correlations and error (Figure S18). The caveat is that O₃ has a much longer atmospheric lifetime than NO₂ so nothing definitive can be concluded by this analysis.**

**To make more conclusive statements about the performance of the model's vertical structure we would need to consider observations from all the ozonesondes (around 100); however, this would require substantial analysis on our end to process these data and accurately represent performance in summary statistics that account for the heterogeneous distribution of observations from different areas, days, and time periods. We agree that this analysis could further support our conclusions; however, this could be a separate study in its own right so we believe that this more detailed investigation is beyond the scope of this study but is worth investigating in the future. We have now modified the text:**

**Before:**

*Although we evaluate the performance of WRF at the surface, we do not consider vertical advection and the vertical mixing scheme in WRF merits further investigation.*

**After:**

*Although we **primarily** evaluate the performance of WRF **meteorology** at the surface, **we also briefly investigate model vertical structure for five ozonesondes from different locations and***

*days (Fig. S14-S18) and find great agreement in temperature and pressure; however, there is more mixed agreement in the ozone mixing ratio, wind speed, and wind direction. Future evaluation of 3D model simulated vertical structure for NO₂ using observations from NASA – such as measurements from the High Spectral Resolution Lidar 2 (HSRL-2) instrument, the Tropospheric Ozone Lidar Network (TolNet), or TRACER-AQ – may be helpful for diagnosing the distinct influences of emissions, meteorology, and chemistry on column NO₂.*

**Before:**

*Additionally, there can be substantial differences in vertical mixing coefficients in different schemes in the models, and these can impact the biases in column concentrations (de Foy et al., 2007; Riess et al., 2023).*

**After:**

*Additionally, there can be substantial differences in vertical mixing coefficients in different schemes in the models, and these can impact the biases in column concentrations (de Foy et al., 2007; Riess et al., 2023). We briefly compare meteorology and the ozone mixing ratio in the WRF-CAMx simulation to ozonesondes data (https://www-air.larc.nasa.gov/cgi-bin/ArcView/traceraq.2021) and find that while temperature and pressure are captured well, there is variable performance in the vertical structure for ozone mixing ratio, wind speed, and wind direction (Fig. S14-S18).*

**Before:**

*Additionally, we compare simulated hourly NO₂ (Fig. S12) and maximum daily eight-hour average or "MDA8" O₃ (Fig. S13) to observations from seventeen TCEQ continuous air monitoring stations (CAMS) operating in Houston. We find poor performance and a strong negative bias in the simulated surface-level NO₂ (NMB=-59%) and generally good performance in the simulated surface-level MDA8 O₃ (NMB=-2.5%) compared to observations*

**After:**

*Additionally, we compare simulated hourly NO₂ (Fig. S12) and maximum daily eight-hour average or "MDA8" O₃ (Fig. S13) to observations from seventeen TCEQ continuous air monitoring stations (CAMS) operating in Houston. We find poor performance and a strong negative bias in the simulated surface-level NO₂ (NMB=-59%) while simulated surface-level MDA8 O₃ has a much weaker bias (NMB=-2.5%) compared to observations. Comparisons to ozonesondes (Fig. S14-S18) suggest that WRF simulates more aggressive vertical mixing than what is observed; this is consistent with our findings of a stronger negative bias at the surface-level than for the columns as emitted NO₂ at the surface is advected vertically quicker in WRF-CAMx than in reality.*

**Minor Comments:**

Line 170 – Please add the swath size for GCAS.

**We thank you for this comment. We have already mentioned the swath size but not the average pixel size which we have now added to the text:**

**Before:**

*The flight strategy of the aircraft included flying the plane in a 'lawnmower' fashion with flight lines spaced 6.3 km apart, ensuring overlap at flight altitude (FL280) with the instrument field of view of 45 degrees creating one gapless map of NO2 up to three times per flight day.*

**After:**

*The flight strategy of the aircraft included flying the plane in a 'lawnmower' fashion with flight lines spaced 6.3 km apart, ensuring overlap at flight altitude (FL280) with the instrument field of view of 45 degrees creating one gapless map of NO2 up to three times per flight day* ***with an average differential slant column pixel size of 250 m × 250 m.***

Line 237 – Previously I commented "Line 190 – For comparison to TROPOMI, you need to regrid the model to the coarser TROPOMI resolution of 3.5x5.5 km2, otherwise the comparison will certainly look poor." The response is as follows:

"In section 3.4 we have done this already and found generally comparable performance albeit improved correlation when comparing results at a coarser resolution."

Please add a reference to Section 3.4 about this then on line 237, otherwise as a reader I would be thrown off.

**We agree that it would be good to reference Section 3.4 here so that the reader is not thrown off and have now updated the text to reflect this:**

**Before:**

*Spatially, we identify the CAMx grid cell in which each Pandora instrument is located and only consider TROPOMI measurements that were regridded to these grid cells*

**Before:**

*Spatially, we identify the CAMx grid cell in which each Pandora instrument is located and only consider TROPOMI measurements that were regridded to these grid cells.* ***We intercompare GCAS, TROPOMI, and CAMx at this resolution but also compare the three datasets at a coarser resolution (Section 3.4) to account for resolution-dependent errors.***

Line 446 - The authors state that there is a non-systematic difference in wind direction on the order of 20°, but then state two sentences later that there is no apparent systematic bias in meteorology. This is contradictory.

**We apologize for the confusion. In the original text what we were trying to convey is that the model does have some random error (i.e., unsystematic; MAE) but it is not biased (MBE). The original text was confusingly written and did not properly emphasize this point, so we have now modified it to reflect this:**

**Before**

*We generally find good agreement in the WRF simulated meteorology (Table S5-S8 and Fig. S2-11); however, the non-systematic differences in wind direction on the order of 20° would likely degrade correlation between observed and simulated $NO_2$ columns. Given that there is no apparent systematic bias in the meteorology, the negative bias in $NO_2$ columns is likely attributable to an underestimate of $NO_x$ emissions.*

**After**

*We **find that the WRF simulated wind direction ($R^2$=0.76 and MBE=8°), temperature ($R^2$=0.71 and MBE=0.39K), and water vapor mixing ratio ($R^2$=0.86 and MBE=-1.45 g/kg)** (Table S5-S8 and Fig. S2-11) **are generally well correlated and minimally-biased compared to observations**; however, **there are some unsystematic** errors **in wind direction (MAE=26°) and poor correlation in wind speed ($R^2$=0.26) that** would likely degrade correlation between observed and simulated $NO_2$ columns. **While there are errors in the meteorological conditions, the biases at the surface are all small – including minimal bias in the wind speed (MBE=-0.02 m/s) – indicating that** the negative biases in $NO_2$ columns are likely attributable to an underestimate of $NO_x$ emissions; **however, the WRF meteorological performance could partially explain the poor correlation and absolute errors in simulated $NO_2$ columns.***

Also, Tables S5-S6 give statistics for all days and non-cloudy days, but not windy vs. calm.

**We thank you for this comment. While we do not specifically include average statistics for windy vs. calm days this can be easily inferred by looking at the wind speeds in Table S6. We find that on the days with calmer conditions ( < 3 m/s) there is usually poorer corelation ($R^2$=0.07, 0.1, and 0.25) while on windier days (> 4 m/s) there is better correlation ($R^2$=0.5 and 0.32). Model performance is well established as being poorer when simulating calmer conditions than windier conditions (e.g., Yu et al., 2022) which is consistent with our results. We have now added a sentence to specifically discuss windy vs. calm conditions:**

**New Text:**

*We also note that generally, the model performance is stronger on windier days – when speeds exceed 4 m/s ($R^2$=0.5 and 0.32) – than on calmer days – when speeds are below 3 m/s ($R^2$=0.07, 0.1, and 0.25).*

Line 550 – Can you give us some more information about the performance of the scheme used here (YSU?) compared to others?

**We thank you for this comment, we have indicated all the schemes used in Table S1. You are correct that we use YSU. A previous study evaluated the performance of YSU for the same period and domain (Liu et al., 2023). We have now mentioned this in the text:**

**New Text:**

*We note that the YSU scheme used in the WRF-CAMx simulation (Table S1) has been shown to underestimate PBL height in the Houston area during the TRACER-AQ campaign (Liu et al., 2023) which would likely impact the vertical distribution of $NO_2$.*

Line 990 – The authors have discussed errors in meteorology throughout the paper including a bias of 20° in wind direction. I am not convinced errors are "minimal". It would be better to say that: with caveats that there are some errors in meteorology, they are unlikely to fully explain the low NO2 bias in the CAMx column and some of this bias may be attributable to underestimated emissions.

**We are sorry for the confusion; we do want to note that there is not a bias (MBE) of 20° in wind direction but there is an unsystematic error (MAE). We believe that we have now responded to this comment in response to your first major comment and have also included your suggested caveat of our conclusions in the main text:**

**New Text:**

*While there are some errors in the meteorology – notably only a modest correlation between simulated and observed wind speed, albeit with little systematic bias, and mixed capturing of vertical structure compared to ozonesondes observations – these errors are unlikely to fully explain the low bias in simulated $NO_2$.*

**Figures and tables referenced in responses:**

[Figure]

**Figure S14: Comparison between WRF-CAMx simulated meteorology and ozone mixing ratios and ozonesondes observations at 11 am on 9/8 at 29.324° N and 94.552° W (Gulf)**

[Figure]

**Figure S15: Comparison between WRF-CAMx simulated meteorology and ozone mixing ratios and ozonesondes observations at 10am on 9/9 at 29.383° N and 94.831° W (Galveston Bay)**

[Figure]

**Figure S16: Comparison between WRF-CAMx simulated meteorology and ozone mixing ratios and ozonesondes observations at 8am on 9/10 at 29.724° N and 95.339° W (University of Houston)**

[Figure]

**Figure S17: Comparison between WRF-CAMx simulated meteorology and ozone mixing ratios and ozonesondes observations at 9am on 9/11 at 29.67° N and 95.06° W (LaPorte)**

[Figure]

**Figure S18: Comparison between WRF-CAMx simulated meteorology and ozone mixing ratios and ozonesondes observations at 1pm on 9/23 at 29.546° N and 95.53° W (Houston SW Airport)**

**Table S1: WRF physics options and data sources**

| WRF Option | Option Selected |
|---|---|
| Analysis Data | 0.25° GDAS (IC/BCs and analysis nudging on the 36 and 12 km domains) |
| Microphysics | Thompson |
| Longwave Radiation | Rapid Radiative Transfer Model (RRTMG) |
| Shortwave Radiation | RRTMG |
| Surface Layer Physics | Revised MM5 surface layer scheme |
| LSM | Noah |
| PBL scheme | Yonsei University (YSU) |
| Cumulus scheme | Multi-Scale Kain-Fritsch (MSKF) on 36/12 km; none for 4/1.333/0.444 km |

**Table S5: Comparison of observed (OBS) and simulated (WRF) wind direction and associated statistics. Red shading indicates days with limited GCAS observations due to cloud coverage. The statistical measures of mean bias error (MBE), mean absolute error (MAE), and Pearson-R squared ($R^2$) are computed using the astropy circular statistics python module (https://docs.astropy.org/en/stable/stats/circ.html).**

| Date | WRF Dir (°) | OBS Dir (°) | $R^2$ | MBE | MAE | N |
|---|---|---|---|---|---|---|
| 09/01 | 184.0 | 180.0 | 0.04 | -1.0 | 36.0 | 130 |
| 09/03 | 136.0 | 185.0 | -0.04 | -49.0 | 60.0 | 137 |
| 09/08 | 13.0 | 17.0 | 0.57 | -10.0 | 29.0 | 137 |
| 09/09 | 20.0 | 37.0 | 0.78 | -11.0 | 24.0 | 149 |
| 09/10 | 81.0 | 75.0 | 0.38 | 6.0 | 17.0 | 149 |
| 09/11 | 89.0 | 92.0 | 0.68 | -3.0 | 14.0 | 151 |
| 09/23 | 63.0 | 88.0 | 0.3 | -24.0 | 28.0 | 118 |
| 09/24 | 87.0 | 97.0 | 0.47 | -8.0 | 20.0 | 137 |
| 09/25 | 70.0 | 59.0 | 0.57 | 9.0 | 24.0 | 114 |
| 09/26 | 97.0 | 96.0 | 0.8 | 0.0 | 19.0 | 117 |
| All Days | 82.0 | 87.0 | 0.76 | -8.0 | 26.0 | 1339 |
| Not Cloudy Days | 67.0 | 73.0 | 0.73 | -5.0 | 21.0 | 1072 |

**Table S6: Comparison of observed (OBS) and simulated (WRF) wind speed and associated statistics. Red shading indicates days with limited GCAS observations due to cloud coverage. The statistical measures of mean bias error (MBE), mean absolute error (MAE), and Pearson-R squared ($R^2$) are defined at the end of this supplement in section S2.**

| Date | WRF Spd (m/s) | OBS Spd (m/s) | $R^2$ | MBE | MAE | N |
|---|---|---|---|---|---|---|
| 09/01 | 3.47 | 2.93 | 0.07 | 0.55 | 1.51 | 145 |
| 09/03 | 3.46 | 3.17 | 0.0 | 0.29 | 1.8 | 151 |
| 09/08 | 3.46 | 2.79 | 0.1 | 0.68 | 1.25 | 150 |
| 09/09 | 3.7 | 3.76 | 0.23 | -0.06 | 0.94 | 152 |
| 09/10 | 4.6 | 4.69 | 0.5 | -0.09 | 0.98 | 149 |
| 09/11 | 4.11 | 5.04 | 0.32 | -0.93 | 1.35 | 151 |
| 09/23 | 3.14 | 3.2 | 0.2 | -0.06 | 1.17 | 130 |
| 09/24 | 3.32 | 3.77 | 0.41 | -0.45 | 0.96 | 142 |
| 09/25 | 2.73 | 2.66 | 0.25 | 0.07 | 0.94 | 135 |
| 09/26 | 3.1 | 3.29 | 0.36 | -0.2 | 0.99 | 131 |
| All Days | 3.53 | 3.55 | 0.26 | -0.02 | 1.2 | 1436 |
| Not Cloudy Days | 3.55 | 3.68 | 0.37 | -0.13 | 1.08 | 1140 |

**Table S7: Comparison of observed (OBS) and simulated (WRF) temperature and associated statistics. Red shading indicates days with limited GCAS observations due to cloud coverage. The statistical measures of mean bias error (MBE), mean absolute error (MAE), and Pearson-R squared ($R^2$) are defined at the end of this supplement in section S2.**

| Date | WRF Temp (K) | OBS Temp (K) | $R^2$ | MBE | MAE | N |
|---|---|---|---|---|---|---|
| 09/01 | 305.5 | 304.2 | 0.0 | 1.3 | 2.28 | 159 |
| 09/03 | 305.0 | 303.0 | 0.0 | 1.99 | 2.61 | 158 |
| 09/08 | 304.5 | 304.1 | 0.75 | 0.42 | 1.16 | 160 |
| 09/09 | 305.1 | 304.4 | 0.78 | 0.66 | 1.15 | 160 |
| 09/10 | 303.1 | 303.2 | 0.76 | -0.1 | 0.93 | 158 |
| 09/11 | 302.5 | 302.3 | 0.74 | 0.18 | 0.99 | 157 |
| 09/23 | 298.4 | 298.6 | 0.69 | -0.26 | 1.14 | 147 |
| 09/24 | 299.2 | 299.3 | 0.72 | -0.11 | 1.02 | 150 |
| 09/25 | 299.5 | 299.9 | 0.81 | -0.33 | 1.02 | 148 |
| 09/26 | 300.0 | 300.1 | 0.69 | -0.02 | 1.18 | 150 |
| All Days | 302.4 | 302.0 | 0.71 | 0.39 | 1.35 | 1547 |
| Not Cloudy Days | 301.6 | 301.5 | 0.86 | 0.06 | 1.07 | 1230 |

**Table S8: Comparison of observed (OBS) and simulated (WRF) water vapor mixing ratio (WVMR) and associated statistics. Red shading indicates days with limited GCAS observations due to cloud coverage. The statistical measures of mean bias error (MBE), mean absolute error (MAE), and Pearson-R squared ($R^2$) are defined at the end of this supplement in section S2.**

| Date | WRF WVMR (g/kg) | OBS WVMR (g/kg) | $R^2$ | MBE | MAE | N |
|---|---|---|---|---|---|---|
| 09/01 | 17.5 | 19.4 | 0.26 | -1.92 | 1.97 | 99 |
| 09/03 | 17.7 | 18.6 | 0.21 | -0.88 | 1.22 | 98 |
| 09/08 | 12.3 | 12.2 | 0.78 | 0.09 | 1.31 | 100 |
| 09/09 | 10.7 | 13.8 | 0.28 | -3.07 | 3.11 | 100 |
| 09/10 | 9.1 | 10.7 | 0.67 | -1.51 | 1.6 | 98 |
| 09/11 | 10.6 | 12.6 | 0.21 | -2.03 | 2.28 | 97 |
| 09/23 | 6.0 | 7.6 | 0.54 | -1.55 | 1.58 | 97 |
| 09/24 | 7.4 | 8.8 | 0.32 | -1.34 | 1.46 | 100 |
| 09/25 | 8.2 | 9.7 | 0.54 | -1.51 | 1.53 | 98 |
| 09/26 | 10.8 | 11.6 | 0.48 | -0.76 | 0.87 | 100 |
| All Days | 11.1 | 12.5 | 0.85 | -1.45 | 1.69 | 987 |
| Not Cloudy Days | 9.4 | 10.9 | 0.64 | -1.46 | 1.72 | 790 |

[Figure]

**Figure S2: Comparison of observed (OBS) and simulated (WRF) wind at 9am and 1pm CST across sixteen ground-level monitors on September 1, 2021. © OpenStreetMap contributors *2023*. Distributed under the Open Data Commons Open Database License (ODbL) v1.0.**

[Figure]

**Figure S3: Comparison of observed (OBS) and simulated (WRF) wind at 9am and 1pm CST across sixteen ground-level monitors on September 3, 2021. © OpenStreetMap contributors *2023*. Distributed under the Open Data Commons Open Database License (ODbL) v1.0.**

[Figure]

**Figure S4: Comparison of observed (OBS) and simulated (WRF) wind at 9am and 1pm CST across sixteen ground-level monitors on September 8, 2021. © OpenStreetMap contributors *2023*. Distributed under the Open Data Commons Open Database License (ODbL) v1.0.**

[Figure]

**Figure S5: Comparison of observed (OBS) and simulated (WRF) wind at 9am and 1pm CST across sixteen ground-level monitors on September 9, 2021. © OpenStreetMap contributors *2023*. Distributed under the Open Data Commons Open Database License (ODbL) v1.0.**

[Figure]

**Figure S6: Comparison of observed (OBS) and simulated (WRF) wind at 9am and 1pm CST across sixteen ground-level monitors on September 10, 2021. © OpenStreetMap contributors *2023*. Distributed under the Open Data Commons Open Database License (ODbL) v1.0.**

[Figure]

**Figure S7: Comparison of observed (OBS) and simulated (WRF) wind at 9am and 1pm CST across sixteen ground-level monitors on September 11, 2021. © OpenStreetMap contributors *2023*. Distributed under the Open Data Commons Open Database License (ODbL) v1.0.**

[Figure]

**Figure S8: Comparison of observed (OBS) and simulated (WRF) wind at 9am and 1pm CST across sixteen ground-level monitors on September 23, 2021. © OpenStreetMap contributors *2023*. Distributed under the Open Data Commons Open Database License (ODbL) v1.0.**

[Figure]

**Figure S9: Comparison of observed (OBS) and simulated (WRF) wind at 9am and 1pm CST across sixteen ground-level monitors on September 24, 2021. © OpenStreetMap contributors *2023*. Distributed under the Open Data Commons Open Database License (ODbL) v1.0.**

[Figure]

**Figure S10: Comparison of observed (OBS) and simulated (WRF) wind at 9am and 1pm CST across sixteen ground-level monitors on September 25, 2021. © OpenStreetMap contributors *2023*. Distributed under the Open Data Commons Open Database License (ODbL) v1.0.**

[Figure]

**Figure S11: Comparison of observed (OBS) and simulated (WRF) wind at 9am and 1pm CST across sixteen ground-level monitors on September 26, 2021. © OpenStreetMap contributors *2023*. Distributed under the Open Data Commons Open Database License (ODbL) v1.0.**